# STOCHASTIC PROJECTIVE SPLITTING: SOLVING SADDLE-POINT PROBLEMS WITH MULTIPLE REGULARIZERS

## ABSTRACT

We present a new, stochastic variant of the projective splitting (PS) family of algorithms for monotone inclusion problems. It can solve min-max and noncooperative game formulations arising in applications such as robust ML without the convergence issues associated with gradient descent-ascent, the current *de facto* standard approach in ML applications. Our proposal is the first version of PS able to use stochastic gradient oracles. It can solve min-max games while handling multiple constraints and nonsmooth regularizers via projection and proximal operators. Unlike other stochastic splitting methods that can solve such problems, our method does not rely on a product-space reformulation of the original problem. We prove almost-sure convergence of the iterates to the solution and a convergence rate for the expected residual. By working with monotone inclusions rather than variational inequalities, our analysis avoids the drawbacks of measuring convergence through the restricted gap function. We close with numerical experiments on a distributionally robust sparse logistic regression problem.

## 1 INTRODUCTION

The most prominent application of optimization in ML is empirical risk minimization. However, inspired by the success of GANs (Goodfellow et al., 2014). , ML practitioners have developed more complicated min-max and adversarial optimization formulations (Yu et al., 2021; Kuhn et al., 2019; Shafieezadeh-Abadeh et al., 2015; Sinha et al., 2018; Lin et al., 2020; Namkoong & Duchi, 2016; Huang et al., 2017; Wadsworth et al., 2018; Zhang et al., 2018; Edwards & Storkey, 2015; Celis & Keswani, 2019). Solving these multi-player games leads to issues not seen when minimizing a single-player loss function. The competitive nature of a game leads to rotational dynamics that can cause intuitive gradient-based methods to fail to converge (Gidel et al., 2019; Daskalakis et al., 2018; Hsieh et al., 2020).

A mathematical framework underlying both convex optimization and saddle-point problems is the *monotone inclusion problem*; see Ryu & Boyd (2016) for an introduction. Methods developed for monotone inclusions will converge for convex-concave, games as they are explicitly designed to handle such problems' governing dynamics. In recent years, monotone inclusion methods and theory have started to receive attention in the ML community (Diakonikolas, 2020; Liu et al., 2021; Ryu et al., 2020; Pathak & Wainwright, 2020), with a focus on monotone variational inequalities, which form a special case of monotone inclusions (Antonakopoulos et al., 2019; Gidel et al., 2019; Daskalakis et al., 2018; Hsieh et al., 2020; Mertikopoulos et al., 2019).

The most prevalent methods for solving min-max games in ML are variants of *gradient descent-ascent* (GDA). This method alternates between a gradient-descent step for the minimizing player and a gradient-ascent step for the maximizing player. Unfortunately, GDA requires additional assumptions to converge on convex-concave games, and it even fails for some simple 2D bilinear games (Gidel et al., 2019, Prop. 1). While there have been several approaches to modify either GDA (Chavdarova et al., 2021; Grnarova et al., 2021; Balduzzi et al., 2018) or the underlying game objective (Mescheder et al., 2018; Nagarajan & Kolter, 2017; Mescheder et al., 2017) to ensure convergence, this paper instead develops a method for solving monotone inclusions that can naturally handle game dynamics.

Our approach builds upon the recently proposed projective splitting (PS) method with forward steps (Johnstone & Eckstein, 2020b). PS is designed specifically for solving monotone inclusions, thus does not fall prey to the convergence issues that plague GDA, at least for convex-concave games. PS is within the general class of projective splitting methods invented by Eckstein & Svaiter (2008) and developed further in Eckstein & Svaiter (2009); Alotaibi et al. (2014); Combettes & Eckstein (2018); Eckstein (2017); Johnstone & Eckstein (2019; 2021; 2020a). These methods work by creating a separating hyperplane between the current iterate and the solution and then moving closer to the solution by projecting the current iterate onto this hyperplane (see Section 3 for an overview). Other than being able to natively handle game dynamics, the primary advantage of PS is that it *fully splits* problems involving an arbitrary number of regularizers and constraints. "Full splitting" means that the method can handle multiple regularizers and constraints through their respective individual proximal and projection operators, along with the smooth terms via gradients. What makes this useful is that many of the regularizers used in ML have proximal operators that are relatively easy to compute; see for example Parikh & Boyd (2013).

Despite these advantages, the preexisting PS framework has a significant drawback: it requires deterministic gradient oracles. This feature makes it impractical for application to large datasets for which stochastic oracles may be the only feasible option.

**Contributions** The primary contribution of this work is a new projective splitting algorithm that allows for a stochastic gradient oracle. We call the method *stochastic projective splitting* (SPS). Our method "fully splits" the monotone inclusion problem

$$\text{Find } z \in \mathbb{R}^d \ \text{ s.t. } \ 0 \in \sum_{i=1}^n A_i(z) + B(z), \tag{1}$$

where $B$ is monotone and $L$-Lipschitz and each $A_i$ is maximal monotone and typically set valued, usually arising from a constraint or a nonsmooth regularizer in the underlying optimization problem or game; see for example Ryu & Boyd (2016) for definitions. For some example ML applications of (1), see Section 2 and Appendix A. Here, an algorithm that "fully splits" (1) means one whose computational steps each involve only the individual operators $A_1, \ldots, A_n, B$. Ours is the first method that can accomplish full splitting without a *product-space reformulation* that recasts (1) as a two-operator problem on a higher-dimensional space, a tactic whose disadvantages are discussed in Appendix F.7. Our method interrogates the Lipschitz operator $B$ through a stochastic oracle. Previous methods splitting (1) have either required a deterministic oracle for $B$, or have made far more restrictive assumptions on the noise or the operators (Briceño-Arias & Combettes, 2011; Combettes & Pesquet, 2012; Malitsky & Tam, 2020; Bot et al., 2019; Van Dung & Vu, 2021) than we will require below. However, the stochastic methods of Alacaoglu et al. (2021) and Böhm et al. (2020), when combined with a product-space reformulation, can solve (1) when all the $A_i$ are subdifferentials of convex functions; see Section 6.

When moving away from a deterministic gradient oracle in projective splitting, a key difficulty is that the generated hyperplanes do not guarantee separation between the solution and the current point. We solve this issue by relaxing the projection: we only update each iterate in the *direction* of the noisy projection and scale its movement by a decreasing stepsize that allows for control of the stochastic error. Using the framework of *stochastic quasi-Fejér monotonicity* (Combettes & Pesquet, 2015), we prove almost-sure convergence of the final iterate and do not require averaging of the iterates (Theorem 1, Section 5). We also provide a non-asymptotic convergence rate for the approximation residual (Theorem 2, Section 5).

A special case of SPS is the recently-developed Double Stepsize Extragradient Method (DSEG) (Hsieh et al., 2020). When $n = 0$ and therefore only $B$ is present in (1), DSEG and SPS coincide. Thus, our method extends DSEG to allow for regularizers and constraints. Our analysis also provides a new interpretation for DSEG as a special case of projective splitting. Our nonasymptotic convergence rate for SPS also applies to DSEG under no additional assumptions. By contrast, the original convergence rate analysis for DSEG requires either strong monotonicity or an error bound.

We close with numerical experiments on a distributionally robust sparse logistic regression problem. This is a nonsmooth convex-concave min-max problem which can be converted to (1) with $n = 2$ set-valued operators. On this problems class, SPS compares well to the possible alternative splitting methods.

**Non-monotone problems**   The work of Hsieh et al. (2020) included a local convergence analysis for DSEG applied to locally monotone problems. For min-max problems, if the objective is locally convex-concave at a solution and DSEG is initialized in close proximity, then for small enough stepsizes it converges to the solution with high probability. It is possible to extend this result to SPS, along with our convergence rate analysis. This result is beyond the scope of this work, but Appendix J provides a proof sketch.

## 2   BACKGROUND ON MONOTONE INCLUSIONS

Since they are so important to SPS, this section provides some background material regarding monotone inclusions, along with their connections to convex optimization, games, and ML. Appendix G discusses their connections to variational inequalities. For a more thorough treatment, we refer to Bauschke & Combettes (2017). See Appendix A for a longer discussion of the applications of monotone inclusions to ML along with several examples.

**Fundamentals**   Let $f : \mathbb{R}^d \to \mathbb{R} \cup \{\infty\}$ be closed, convex, and proper (CCP). Recall that its *subdifferential* $\partial f$ is given by $\partial f(x) \doteq \{g : f(y) \geq f(x) + g^\top (y - x)\}$. The map $\partial f$ has the property

$$u \in \partial f(x), v \in \partial f(y) \implies (u - v)^\top (x - y) \geq 0,$$

and any point-to-set map having this property is called a *monotone operator*. A monotone operator $T$ is called *maximal* if no additional points can be included in the image $T(x)$ of any $x \in \mathbb{R}^d$ without violating the above property (Bauschke & Combettes, 2017, Def. 20.20). Subgradient maps of CCP functions are maximal (Bauschke & Combettes, 2017, Thm. 20.25). A minimizer of $f$ is any $x^*$ such that $0 \in \partial f(x^*)$. This is perhaps the simplest example of a *monotone inclusion*, the problem of finding $x$ such that $0 \in T(x)$, where $T$ is a monotone operator. If $f$ is smooth, then $\partial f(x) = \{\nabla f(x)\}$ for all $x$, and the monotone inclusion $0 \in \partial f(x)$ is equivalent to the first-order optimality condition $0 = \nabla f(x)$.

Under certain regularity conditions  (Bauschke & Combettes, 2017, Cor. 16.5), minimizing a sum of CCP functions $f_1, \ldots, f_n$ is equivalent to solving the monotone inclusion formed from the sum of their subdifferentials:

$$x^* \in \arg\min_{x \in \mathbb{R}^d} \sum_{i=1}^n f_i(x) \iff 0 \in \sum_{i=1}^n \partial f_i(x^*). \tag{2}$$

As throughout this paper for all set addition operations, the summation on the right-hand side of (2) is the *Minkowski sum* $\sum_{i=1}^n S_i = \{\sum_{i=1}^n s_i \mid s_i \in S_i \ \forall \, i \in 1..n\}$. For a convex set $X$, a constraint $x \in C$ for some convex set $C$ may be imposed by setting one of the $f_i$ to be the *indicator function* $\iota_C$, defined by $\iota_C(x) = 0$ for $x \in C$ and $\iota_C(x) = +\infty$ for $x \notin C$. Indicator functions of closed convex sets are CCP (Bauschke & Combettes, 2017, Ex. 1.25), and the subgradient map of $\iota_C$ is also referred to as the *normal cone map* $N_C$ of $C$ (Bauschke & Combettes, 2017, Def. 6.37). Multiple constraints may be imposed by including multiple indicator functions in (2).

**ML applications**   The form (2) can be used to model ML problems with multiple constraints and/or nonsmooth regularizers, including sparse and overlapping group lasso (Jacob et al., 2009), sparse and low-rank matrix estimation problems (Richard et al., 2012), and rare feature selection (Yan & Bien, 2020); see Pedregosa & Gidel (2018) for an overview.

**Games**   Consider a two-player noncooperative game in which each player tries to selfishly minimize its own loss, with each loss depending on the actions of both players. Typically, the goal is to find a Nash equilibrium, in which neither player can improve its loss by changing strategy:

$$x^* \in \arg\min_{x \in \Theta} F(x, y^*) \quad \text{and} \quad y^* \in \arg\min_{y \in \Omega} G(x^*, y). \tag{3}$$

Assuming that the admissible strategy sets $\Theta \subseteq \mathbb{R}^{d_x}$ and $\Omega \subseteq \mathbb{R}^{d_y}$ are closed and convex and that $F$ and $G$ are differentiable, then writing the first-order necessary conditions for each optimization problem in (3) yields

$$0 \in \left[ \begin{array}{c} \nabla_x F(x^*, y^*) \\ \nabla_y G(x^*, y^*) \end{array} \right] + \left( N_\Theta(x^*) \times N_\Omega(y^*) \right). \tag{4}$$

If $G = -F$, then (3) is a min-max game. If $F$ is also convex in $x$ and concave in $y$, then $B : (x, y) \mapsto (\nabla_x F(x, y), -\nabla_y F(x, y))^\top$ is monotone[1] on $\mathbb{R}^{d_x + d_y}$ (Rockafellar, 1970). In many applications, $B$ is also Lipschitz continuous. In this situation, (4) is a monotone inclusion involving two operators $B$ and $N_{\Theta \times \Omega}$, with $B$ being Lipschitz. Using the simultaneous version of GDA on (3) is equivalent to applying the forward-backward method (FB) (Bauschke & Combettes, 2017, Thm. 26.14) to (4). However, convergence of FB requires that the operator $B$ be *cocoercive* (Bauschke & Combettes, 2017, Def. 4.10), and not merely Lipschitz (Bauschke & Combettes, 2017, Thm. 26.14). Thus, simultaneous GDA fails to converge for (3) without additional assumptions; see Gidel et al. (2019, Prop. 1) for a simple counterexample.

Regularizers and further constraints may be imposed by adding more operators to (4). For example, if one wished to apply a (nonsmooth) convex regularizer $r : \mathbb{R}^{d_x} \to \mathbb{R} \cup \{+\infty\}$ to the $x$ variables and a similar regularizer $d : \mathbb{R}^{d_y} \to \mathbb{R} \cup \{+\infty\}$ to the $y$ variables, one would add the operator $A_2 : (x, y) \mapsto \partial r(x) \times \partial d(y)$ to the right-hand side of (4).

**ML applications of games**   Distributionally robust supervised learning (DRSL) is an emerging framework for improving the stability and reliability of ML models in the face of distributional shifts (Yu et al., 2021; Kuhn et al., 2019; Shafieezadeh-Abadeh et al., 2015; Sinha et al., 2018; Lin et al., 2020; Namkoong & Duchi, 2016). Common approaches to DRSL formulate the problem as a min-max game between a learner selecting the model parameters and an adversary selecting a worst-case distribution subject to some ambiguity set around the observed empirical distribution. This min-max problem is often further reduced to either a finite-dimensional saddlepoint problem or a convex optimization problem.

DRSL is a source of games with multiple constraints/regularizers. One such formulation, based on Yu et al. (2021), is discussed in the experiments below. The work in Namkoong & Duchi (2016) uses an ambiguity set based on $f$-divergences, while Sinha et al. (2018) introduce a Lagrangian relaxation of the Wasserstein ball. When applied to models utilizing multiple regularizers (Jacob et al., 2009; Richard et al., 2012; Yan & Bien, 2020), both of these approaches lead to min-max problems with multiple regularizers.

Other applications of games in ML, although typically nonconvex, include generative adversarial networks (GANs) (Goodfellow et al., 2014; Arjovsky et al., 2017; Loizou et al., 2020; 2021; Mishchenko et al., 2020), fair classification (Wadsworth et al., 2018; Zhang et al., 2018; Edwards & Storkey, 2015; Celis & Keswani, 2019), and adversarial privacy (Huang et al., 2017).

**Resolvents, proximal operators, and projections**   A fundamental computational primitive for solving monotone inclusions is the *resolvent*. The resolvent of a monotone operator $A$ is defined to be $J_A \doteq (I + A)^{-1}$, where $I$ is the identity operator and the inverse of any operator $T$ is simply $T^{-1} : x \mapsto \{y : Ty \ni x\}$. If $A$ is maximal monotone, then for any $\rho > 0$, $J_{\rho A}$ is single valued, nonexpansive, and has domain equal to $\mathbb{R}^d$ (Bauschke & Combettes, 2017, Thm. 21.1 and Prop. 23.8). Resolvents generalize proximal operators of convex functions: the proximal operator of a CCP function $f$ is

$$\text{prox}_{\rho f}(t) \doteq \underset{x \in \mathbb{R}^d}{\arg\min} \left\{ \rho f(x) + (1/2)\|x - t\|^2 \right\}.$$

It is easily proved that $\text{prox}_{\rho f} = J_{\rho \partial f}$. Like proximal operators, resolvents generalize projection onto convex sets: if $f = \iota_{\mathcal{C}}$, then $J_{\rho N_C} = \text{prox}_{\rho f} = \text{proj}_{\mathcal{C}}$ for any $\rho > 0$. In many ML applications, proximal operators, and hence resolvents, are relatively straightforward to compute. For examples, see Parikh & Boyd (2013, Sec. 6).

**Operator splitting methods**   *Operator splitting methods* attempt to solve monotone inclusions such as (1) by a sequence of operations that each involve only one of the operators $A_1, \ldots, A_n, B$. Such methods are often presented in the context of convex optimization problems like (2), but typically apply more generally to monotone inclusions such as (1). In the specific context of (1), each iteration of such a method ideally handles each $A_i$ via its resolvent and the Lipschitz operator $B$ by explicit (not stochastic) evaluation. This is a feasible approach if the original problem can be decomposed in

---

[1]Sufficient conditions for the monotonicity of (4) in the case where $G \neq -F$ are discussed in *e.g.* Scutari et al. (2014); Briceño-Arias & Combettes (2013).

such a way that the resolvents of each $A_i$ are relatively inexpensive to compute, and full evaluations of $B$ are possible. Although not discussed here, more general formulations in which matrices couple the arguments of the operators can broaden the applicability of operator splitting methods.

## 3    THE PROJECTIVE SPLITTING FRAMEWORK

Before introducing our proposed method, we give a brief introduction to the projective splitting class of methods.

**The extended solution set**    Projective splitting is a primal-dual framework and operates in an extended space of primal and dual variables. Rather than directly finding a solution to (1), we find a point in the *extended solution set* (or *Kuhn-Tucker set*)

$$\mathcal{S} \doteq \left\{ (z, w_1, \ldots, w_{n+1}) \mid w_i \in A_i(z) \,\forall\, i \in 1..n, w_{n+1} = B(z), \sum_{i=1}^{n+1} w_i = 0 \right\}. \tag{5}$$

Given $p^* = (z^*, w_1^* \ldots, w_{n+1}^*) \in \mathcal{S}$, it is straightforward to see that $z^*$ solves (1). Conversely, given a solution $z^*$ to (1), there must exist $w_1^*, \ldots, w_{n+1}^*$ such that $(z^*, w_1^*, \ldots, w_{n+1}^*) \in \mathcal{S}$. Suppose $p^* = (z^*, w_1^* \ldots, w_{n+1}^*) \in \mathcal{S}$. Since $z^*$ solves (1), $z^*$ is typically referred to as a *primal solution*. The vectors $w_1^*, \ldots, w_{n+1}^*$ solve a dual inclusion not described here, and are therefore called a *dual solution*. It can be shown that $\mathcal{S}$ is closed and convex; see for example Johnstone & Eckstein (2020b). We will assume throughout that a solution to (1) exists, therefore the set $\mathcal{S}$ is nonempty.

**Separator-projection framework**    Projective splitting methods are instances of the general *separator-projection* algorithmic framework for locating a member of a closed convex set $\mathcal{S}$ within a linear space $\mathcal{P}$. Each iteration $k$ of algorithms drawn from this framework operates by finding a set $H_k$ that separates the current iterate $p^k \in \mathcal{P}$ from $\mathcal{S}$, meaning that $\mathcal{S}$ is entirely in the set and $p^k$ typically is not. One then attempts to "move closer" to $\mathcal{S}$ by projecting the $p^k$ onto $H_k$. In the particular case of projective splitting applied to the problem (1) using (5), we select the space $\mathcal{P}$ to be

$$\mathcal{P} \doteq \left\{ (z, w_1, \ldots, w_{n+1}) \in \mathbb{R}^{(n+2)d} \mid \sum_{i=1}^{n+1} w_i = 0 \right\}, \tag{6}$$

and each separating set $H_k$ to be the half space $\{p \in \mathcal{P} \mid \varphi_k(p) \le 0\}$ generated by an affine function $\varphi_k : \mathcal{P} \to \mathbb{R}$. The general intention is to construct $\varphi_k$ such that $\varphi_k(p^k) > 0$, but $\varphi_k(p^*) \le 0$ for all $p^* \in \mathcal{S}$. The construction employed for $\varphi_k$ in the case of (1) and (5) is of the form

$$\varphi_k(z, w_1, \ldots, w_{n+1}) \doteq \sum_{i=1}^{n+1} \langle z - x_i^k, y_i^k - w_i \rangle \tag{7}$$

for some points $(x_i^k, y_i^k) \in \mathbb{R}^{2d}$, $i \in 1..(n+1)$, that must be carefully chosen (see below). Any function of the form (7) can be shown to be affine when restricted to $\mathcal{P}$. As mentioned above, the standard separator-projection algorithm obtains its next iterate $p^{k+1}$ by projecting $p^k$ onto $H_k$. This calculation involves the usual projection step for a half space, namely

$$p^{k+1} = p^k - \alpha_k \nabla \varphi_k, \quad \text{where} \quad \alpha_k = \varphi_k(p^k)/\|\nabla \varphi_k\|^2, \tag{8}$$

and the gradient $\nabla \varphi_k$ is computed relative to $\mathcal{P}$, thus resulting in $p^{k+1} \in \mathcal{P}$, i.e. $\nabla \varphi_k = \left( \sum_{i=1}^{n+1} y_i^k, x_1^k - \bar{x}^k, \ldots, x_{n+1}^k - \bar{x}^k \right)$ where $\bar{x}^k = \frac{1}{n+1} \sum_{i=1}^{n+1} x_i^k$.

## 4    PROPOSED METHOD

The proposed method is given in Algorithm 1 and called *Stochastic Projective Splitting* (SPS). Unlike prior versions of projective splitting, SPS does not employ the stepsize $\alpha_k$ of (8) that places the next iterate exactly on the hyperplane given by $\varphi_k(p) = 0$. Instead, it simply moves in the direction $-\nabla \varphi_k$ with a pre-defined stepsize $\{\alpha_k\}$. This fundamental change is required to deal with the stochastic noise on lines 6 and 8. This noise could lead to the usual choice of $\alpha_k$ defined in (8) being unstable and difficult to analyze. In order to guarantee convergence, the parameters $\alpha_k$ and $\rho_k$ must be chosen to satisfy certain conditions given below. Note that the gradient is calculated with respect to the subspace $\mathcal{P}$ defined in (6); since the algorithm is initialized within $\mathcal{P}$, it remains in $\mathcal{P}$, within which $\varphi_k$ is affine. Collectively, the updates on lines 9-10 are equivalent to $p^{k+1} = p^k - \alpha_k \nabla \varphi_k$, where $p^k = (z^k, w_1^k, \ldots, w_{n+1}^k)$.

Note that SPS does not explicitly evaluate $\varphi_k$, which is only used in the analysis, but it does keep track of $(x_i^k, y_i^k)$ for $i \in 1..(n+1)$. The algorithm's memory requirements scale linearly with the number of nonsmooth operators $n$ in the inclusion (1), with the simplest implementation storing $(3n+5)d$ working-vector elements. This requirement can be reduced to $(n+7)d$ through a technique discussed in Appendix H. In most applications, $n$ will be small, for example 2 or 3.

**Updating** $(x_i^k, y_i^k)$    The variables $(x_i^k, y_i^k)$ are updated on lines 3-8 of Algorithm 1, in which $e^k$ and $\epsilon^k$ are $\mathbb{R}^d$-valued random variables defined on a probability space $(\Omega, \mathcal{F}, P)$. For $B$ we use a new, noisy version of the two-forward-step procedure from Johnstone & Eckstein (2020b). For each $A_i$, $i \in 1..n$, we use the same resolvent step used in previous projective splitting papers, originating with (Eckstein & Svaiter, 2008). In the case $\epsilon^k = e^k = 0$, the selection of the $(x_i^k, y_i^k)$ is identical to that proposed by Johnstone & Eckstein (2020b), resulting in the hyperplane $\{p : \varphi_k(p) = 0\}$ strictly separating $p^k$ from $\mathcal{S}$.

SPS achieves full splitting of (1): each $A_i$ is processed separately using a resolvent and the Lipschitz term $B$ is processed via a stochastic gradient oracle. When the $A_i$ arise from regularizers or constraints, as discussed in Section 2, their resolvents can be readily computed so long as their respective proximal/projection operators have a convenient form.

**Noise assumptions**    Let $\mathcal{F}_k \doteq \sigma(p^1, \ldots, p^k)$ and $\mathcal{E}_k \doteq \sigma(\epsilon^k)$. The stochastic estimators for the gradients, $r^k$ and $y_{n+1}^k$, are assumed to be unbiased, that is, the noise terms have mean 0 conditioned on the past:

$$\mathbb{E}[\epsilon^k | \mathcal{F}_k] = 0, \quad \mathbb{E}[e^k | \mathcal{F}_k] = 0 \quad a.s. \tag{9}$$

We impose the following mild assumptions on the variance of the noise:

$$\mathbb{E}\left[\|\epsilon^k\|^2 | \mathcal{F}_k\right] \leq N_1 + N_2 \|B(z^k)\|^2 \quad a.s. \tag{10}$$

$$\mathbb{E}\left[\|e^k\|^2 | \mathcal{F}_k, \mathcal{E}_k\right] \leq N_3 + N_4 \|B(x_{n+1}^k)\|^2 \quad a.s., \tag{11}$$

where $0 \leq N_1, N_2, N_3, N_4 < \infty$. We do not require $e^k$ and $\epsilon^k$ to be independent of one another.

**Stepsize choices**    The stepsizes $\rho_k$ and $\alpha_k$ are assumed to be deterministic. A constant stepsize choice which attains a non-asymptotic convergence rate will be considered in the next section (Theorem 2). The stepsize conditions we will impose to guarantee almost-sure convergence (Theorem 1) are

$$\sum_{k=1}^{\infty} \alpha_k \rho_k = \infty, \quad \sum_{k=1}^{\infty} \alpha_k^2 < \infty, \quad \sum_{k=1}^{\infty} \alpha_k \rho_k^2 < \infty, \text{ and } \rho_k \leq \overline{\rho} < 1/L. \tag{12}$$

For example, in the case $L = 1$, a particular choice which satisfies these constraints is

$$\alpha_k = k^{-0.5-p} \text{ for } 0 < p < 0.5, \text{ and } \rho_k = k^{-0.5+t} \text{ for } p \leq t < 0.5p + 0.25.$$

For simplicity, the stepsizes $\tau$ used for the resolvent updates in lines 3-5 are fixed, but they could be allowed to vary with both $i$ and $k$ so long as they have finite positive lower and upper bounds.

---

**Algorithm 1:** Stochastic Projective Splitting (SPS)

---

**Input :** $p^1 = (z^1, w_1^1, \ldots, w_{n+1}^1)$ s.t. $\sum_{i=1}^{n+1} w_i^1 = 0$, $\{\alpha_k, \rho_k\}_{k=1}^{\infty}$, $\tau > 0$

1 **for** $k = 1, 2, \ldots$ **do**
2      **for** $i \in 1..n$ **do**
3          $t_i^k = z^k + \tau w_i^k$
4          $x_i^k = J_{\tau A_i}(t_i^k)$
5          $y_i^k = \tau^{-1}(t_i^k - x_i^k)$
6      $r^k = B(z^k) + \epsilon^k$             `// ` $\epsilon^k$ ` is unknown noise term`
7      $x_{n+1}^k = z^k - \rho_k(r^k - w_{n+1}^k)$
8      $y_{n+1}^k = B(x_{n+1}^k) + e^k$         `// ` $e^k$ ` is unknown noise term`
9      $z^{k+1} = z^k - \alpha_k \sum_{i=1}^{n+1} y_i^k$
10     $w_i^{k+1} = w_i^k - \alpha_k(x_i^k - \frac{1}{n+1} \sum_{i=1}^{n+1} x_i^k) \quad i \in 1..(n+1)$

---

## 5  MAIN THEORETICAL RESULTS

**Theorem 1.** *Suppose $A_1, \ldots, A_n$ are maximal monotone, $B$ is $L$-Lipschitz and monotone, and a solution to* (1) *exists. For Algorithm 1, suppose* (9)-(12) *hold. Then with probability one it holds that $z^k \to z^*$, where $z^*$ solves* (1). *Further, with probability one, $x_i^k \to z^*$ for $i = 1, \ldots, n$.*

**Proof sketch**   Theorem 1 is proved in Appendix C, but we provide a brief sketch here. The proof begins by deriving a simple recursion inspired by the analysis of SGD (Robbins & Monro, 1951). Since $p^{k+1} = p^k - \alpha_k \nabla \varphi_k$, a step of projective splitting can be viewed as GD applied to the affine hyperplane generator function $\varphi_k$. Thus, for any $p^* \in \mathcal{P}$,

$$\|p^{k+1} - p^*\|^2 = \|p^k - p^*\|^2 - 2\alpha_k \langle \nabla \varphi_k, p^k - p^* \rangle + \alpha_k^2 \|\nabla \varphi_k\|^2$$
$$= \|p^k - p^*\|^2 - 2\alpha_k(\varphi_k(p^k) - \varphi_k(p^*)) + \alpha_k^2 \|\nabla \varphi_k\|^2, \qquad (13)$$

where in the second equation we have used that $\varphi_k(p)$ is affine on $\mathcal{P}$. The basic strategy is to show that, for any $p^* \in \mathcal{S}$,

$$\mathbb{E}[\|\nabla \varphi_k\|^2 | \mathcal{F}_k] \leq C_1 \|p^k - p^*\|^2 + C_2 \quad a.s.$$

for some $C_1, C_2 > 0$. This condition allows one to establish stochastic quasi-Fejér monotonicity (SQFM) (Combettes & Pesquet, 2015, Proposition 2.3) of the iterates to $\mathcal{S}$. One consequence of SQFM is that with probability one there exists a subsequence $v_k$ such that $\varphi_{v_k}(p^{v_k}) - \varphi_{v_k}(p^*)$ converges to 0. Furthermore, roughly speaking, we show that $\varphi_k(p^k) - \varphi_k(p^*)$ provides an upper bound on the following "approximation residual" for SPS:

$$G_k \doteq \sum_{i=1}^n \|y_i^k - w_i^k\|^2 + \sum_{i=1}^n \|z^k - x_i^k\|^2 + \|B(z^k) - w_{n+1}^k\|^2. \qquad (14)$$

$G_k$ provides an approximation error for SPS, as formalized in the following lemma:

**Lemma 1.** *For SPS, $p^k = (z^k, w_1^k, \ldots, w_{n+1}^k) \in \mathcal{S}$ if and only if $G_k = 0$.*

Since $y_i^k \in A_i(x_i^k)$ for $i \in 1..n$, having $G_k = 0$ implies that $z^k = x_i^k$, $w_i^k = y_i^k$, and thus $w_i^k \in A_i(z^k)$ for $i \in 1..n$. Since $w_{n+1}^k = B(z^k)$ and $\sum_{i=1}^{n+1} w_i^k = 0$, it follows that $z^k$ solves (1). The reverse direction is proved in Appendix D.

The quantity $G_k$ generalizes the role played by the norm of the gradient in algorithms for smooth optimization. In particular, in the special case where $n = 0$ and $B(z) = \nabla f(z)$ for some smooth convex function $f$, one has $G_k = \|\nabla f(z^k)\|^2$.

Combining the properties of $G_k$ with other results following from SQFM (such as boundedness) will allow us to derive almost-sure convergence of the iterates to a solution of (1).

**Convergence rate**   We can also establish non-asymptotic convergence rates for the approximation residual $G_k$:

**Theorem 2.** *Fix the total iterations $K \geq 1$ of Algorithm 1 and set*

$$\forall k = 1, \ldots, K : \rho_k = \rho \doteq \min\left\{K^{-1/4}, 1/2L\right\} \quad and \quad \alpha_k = C_f \rho^2 \qquad (15)$$

*for some $C_f > 0$. Suppose* (9)-(11) *hold. Then*

$$(1/K)\sum_{j=1}^K \mathbb{E}[G_j] = \mathcal{O}(K^{-1/4})$$

*where the constants are given (along with the proof) in Appendix E.*

Theorem 2 implies that if we pick an iterate $J$ uniformly at random from $1..K$, then the expected value of $G_J$ is $\mathcal{O}(K^{-1/4})$. As far as we know, this is the first convergence rate for a stochastic full-splitting method solving (1) in the general discontinuous (i.e. set-valued) monotone inclusion case, and it is not clear whether it can be improved, either by a better analysis or a better method. Faster rates are certainly possible for deterministic methods under various continuity assumptions; Tseng's method obtains $\mathcal{O}(K^{-1})$ rate (Monteiro & Svaiter, 2010) and the accelerated Halpern iteration under Lipschitz continuity obtains $\mathcal{O}(K^{-2})$ rate (Diakonikolas, 2020). While our rate may seem slow, it is worth remembering that (1) features $n$ discontinuous operators $A_i$, so we expect rates at least as slow as nonsmooth convex optimization, but perhaps worse because (1) is far more general than convex optimization. For a different error metric, the restricted gap function, in the special case of variational inequalities, faster rates have been established in Juditsky et al. (2011) and Böhm et al. (2020). However, it is unclear how to relate the restricted gap function to $G_k$, so these rates may not be directly comparable to Theorem 2.

## 6 RELATED WORK

Arguably the three most popular classes of operator splitting algorithms are forward-backward splitting (FB) (Combettes & Pesquet, 2011), Douglas-Rachford splitting (DR) (Lions & Mercier, 1979), and Tseng's method (Tseng, 2000). The extragradient method (EG) is similar to Tseng's method, but has more projection steps per iteration and only applies to variational inequalities (Korpelevich, 1977; Nemirovski, 2004; Li et al., 2021). The popular Alternating Direction Method of Multipliers (ADMM), in its standard form, is a dual application of DR (Gabay, 1983). The three-operator splitting method (Davis & Yin, 2017) can only be applied to (1) if $B$ is cocoercive rather than merely Lipchitz, and thus its usefulness is mostly limited to optimization applications and not games. FB, DR, and Tseng's method apply to monotone inclusions involving two operators, with varying assumptions on one of the operators. It is possible to derive splitting methods for the more complicated inclusion (1), involving more than two operators, by applying an appropriate 2-operator splitting method such as Tseng's method to a product-space reformulation (PSR) (Briceño-Arias & Combettes, 2011; Combettes & Pesquet, 2012) (for more on PSR, see Appendix F). The recently developed forward-reflected-backward (FRB) method (Malitsky & Tam, 2020) can be used in the same way. However, there are several disadvantages to using a PSR, as discussed in Appendix F.7.

By using a PSR, the stochastic methods of Alacaoglu et al. (2021) and Böhm et al. (2020) can be applied to (1) in the case that each $A_i$ is a subdifferential. Both of these methods are analyzed in terms of the *restricted gap function*. This merit function has a drawback compared with our approximation residual in that it requires one to find a bound for the iterates. However, Alacaoglu et al. (2021) and Böhm et al. (2020) do not provide such a bound, meaning that their convergence rate results are somewhat incomplete. We discuss this issue in Appendix G.

Theoretical convergence of the method of Böhm et al. (2020) requires the use of averaging, since the final iterate does not converge for certain problems (Hsieh et al., 2020). Empirically, averaging tends to be slow and to destroy regularizer-induced structural properties such as sparsity or low matrix rank, so its utility is largely theoretical and it is usually avoided in practice. Furthermore, averaging loses even its theoretical benefits for nonconvex problems, so its use in such cases is rarer still. Another drawback of the analysis of Böhm et al. (2020) is that, unlike in SPS, the resolvent (proximal) stepsizes also need to vanish.

The method of Alacaoglu et al. (2021) applies variance reduction techniques to FRB. It only applies to finite-sum problems and requires the periodic computation of a full batch gradient, making it somewhat less flexible and scalable than our method. On the other hand, it has an accelerated ergodic rate for the restricted gap function in the variational inequality setting. We compare the empirical performance of SPS with Alacaoglu et al. (2021), Böhm et al. (2020), and several deterministic methods using PSR in the numerical experiments described in Section 7.

Additional related work is discussed in Appendix B.

## 7 EXPERIMENTS

We now present some numerical results on distributionally robust supervised learning (DRSL) problems. We follow the approach of Yu et al. (2021), which introduced a min-max formulation of Wasserstein DRSL. While other approaches reduce the problem to convex optimization, Yu et al. (2021) reduce it to a finite-dimensional min-max problem amenable to the use of stochastic methods on large datasets. However, unlike our proposed SPS method, the variance-reduced extragradient method that Yu et al. (2021) propose cannot handle multiple nonsmooth regularizers or constraints on the model parameters. Consequently, we consider distributionally robust sparse logistic regression (DRSLR), a problem class equivalent to that considered in Yu et al. (2021), but with an added $\ell_1$ regularizer, a standard tool to induce sparsity. See the Appendix I for the full problem definition.

We compared our SPS method to several methods for solving DRSLR for a collection of real datasets from the LIBSVM repository (Chang & Lin, 2011). We implemented SPS with $\alpha_k = C_d k^{-0.51}$ and $\rho_k = C_d k^{-0.25}$ and called it *SPS-decay*. We also implement SPS with the fixed stepsize given in (15) and called it *SPS-fixed*. We compared the method to deterministic projective splitting (Johnstone & Eckstein, 2020b) and the following methods based on PSR: Tseng's method (Tseng, 2000; Combettes & Pesquet, 2012), the forward-reflected-backward (FRB) method (Malitsky & Tam, 2020), the

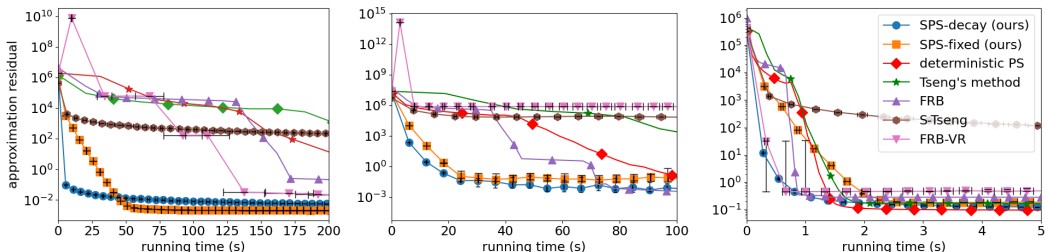

Figure 1: Approximation residual versus running time for three LIBSVM benchmark datasets, with the markers at 10-iteration intervals. Left: epsilon, middle: SUSY, right: real-sim. For the stochastic algorithms (SPS, S-Tseng, and FRB-VR), we plot the median results over 10 trials, with unit standard deviation horizontal error bars for the running time and the vertical error bars displaying the min-to-max range of the approximation residual. The code is provided in the supplementary material.

stochastic Tseng (S-Tseng) method of Böhm et al. (2020), and the variance-reduced stochastic FRB method (Alacaoglu et al., 2021), abbreviated FRB-VR. The S-Tseng and FRB-VR algorithms appear to be the only stochastic splitting methods other than SPS applicable to the tested problem class.

Figure 1 show results for three LIBSVM standard datasets: *epsilon*[2] ($m = 4 \cdot 10^5$, $d = 2000$), *SUSY* (Baldi et al., 2014; Dua & Graff, 2017) ($m = 2 \cdot 10^6$, $d = 18$), and *real-sim*[3] ($m = 72{,}309$, $d = 20{,}958$).

To measure the progress of the algorithms, we used the "approximation residual" $R_k$ defined in Appendix F. As with $G_k$, having $R_k = 0$ implies that $z^k$ solves (1). We use $R_k$ instead of $G_k$ because it is also possible to compute essentially the same measure of convergence from the iterates of the other tested algorithms, establishing a fair comparison. Appendix F provides the details of the derivation of the residual measure for each algorithm, explores the relationship between $R_k$ and $G_k$, and provides additional implementation details.

Figure 1 plots the approximation residual versus running time for all seven algorithms under consideration. The computations were performed using Python 3.8.3 and `numpy` on a 2019 MacBook Pro with a 2.4GHz 8-core Intel I9 processor and 32GB of RAM . Being a stochastic method, SPS-decay seems to outperform the deterministic methods at obtaining a medium-accuracy solution quickly. It also seems to outperform the stochastic PSR-based methods S-Tseng and FRB-VR.

## 8 CONCLUSIONS AND FUTURE WORK

We have developed and analyzed a stochastic splitting method that can handle min-max problems with multiple regularizers and constraints. Going forward, this development should make it possible to incorporate regularizers and constraints into adversarial formulations trained from large datasets.

Recent versions of deterministic projective splitting (Combettes & Eckstein, 2018; Johnstone & Eckstein, 2020b) allow for asynchronous and incremental operation, meaning that not all operators need to be activated at every iteration, with some calculations proceeding with stale inputs. Such characteristics make projective splitting well-suited to distributed implementations. Many of our SPS results may be extended to allow for these variations, but we leave those extensions to future work.

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

## A  ML APPLICATIONS OF THE MONOTONE INCLUSION (1)

There are two main classes of applications of (1) in ML: optimization problems and saddle-point games.

**Optimization Problems**  In this case the monotone inclusion arises from finding the zero of a sum of subgradients of convex functions, as discussed in Section 2. It is typical in ML to solve the empirical risk minimization problem

$$\min_{x\in\mathbb{R}^d} \frac{1}{m}\sum_{j=1}^{m} f_j(x) + \sum_{i=1}^{n} r_i(x) \tag{16}$$

over a size-$m$ dataset. Usually, the gradient of the loss function $f_j$ for each datapoint $j$ is Lipschitz continuous. The terms $r_i$ may be regularizers used to reduce overfitting or encourage structural properties such as sparsity or low matrix rank. They also may represent constraints on the parameters such as nonnegativity or the being in the probability simplex. Crucially, these regularizers are rarely differentiable. The first-order necessary condition for the solution of (16) is

$$0 \in \nabla f(x^*) + \sum_{i=1}^{n} \partial r_i(x^*), \tag{17}$$

where $f(x) \doteq \frac{1}{m} \sum_{j=1}^{m} f_j(x)$, thus $\nabla f(x) \doteq \frac{1}{m} \sum_{j=1}^{m} \nabla f_j(x)$. The inclusion (17) is a special case of (1), and our method may use the standard stochastic oracle for $\nabla f(x)$, namely

$$\frac{1}{|\mathbf{B}|} \sum_{j \in \mathbf{B}} \nabla f_j(z)$$

which subsamples a randomly selected minibatch of datapoints $\mathbf{B} \in \{1, \ldots, m\}$.

**Games**   Consider the following nonsmooth Nash equilibrium problem

$$x^* \in \arg\min_{x \in \mathbb{R}^{d_x}} F(x, y^*) + \sum_{i=1}^{n_1} r_i(x) \quad \text{and} \quad y^* \in \arg\min_{y \in \mathbb{R}^{d_y}} G(x^*, y) + \sum_{i=1}^{n_2} d_i(y). \quad (18)$$

The terms $\sum_{i=1}^{n_1} r_i(x)$ and $\sum_{i=1}^{n_2} d_i(y)$ once again represent regularizers and constraints on each player's strategy. Note that min-max (saddle-point) problems correspond to having $F(x, y) = -G(x, y)$. Under appropriate convexity conditions and constraint qualifications, the solutions of (18) correspond to the solutions of the following monotone inclusion in the form of (1):

$$0 \in \begin{bmatrix} \nabla_x F(x^*, y^*) \\ \nabla_y G(x^*, y^*) \end{bmatrix} + \sum_{i=1}^{\max\{n_1, n_2\}} \left( \partial r_i(x^*) \times \partial d_i(y^*) \right) \quad (19)$$

where for $i > \min\{n_1, n_2\}$ we include "dummy functions", either $r_i(x) = 0$ when $n_1 < n_2$ or $d_i(y) = 0$ when $n_1 < n_2$. If the functions $F$ and $G$ arise as averages in the same we as $f$ in (16), then our method may again use a stochastic oracle for them.

**Distributionally-Robust ML**   One example application of (19) is distributionally-robust ML, as demonstrated in the numerical experiment in Section 7. The full problem statement is given in Appendix I.

**Lagrangian Duality**   Another application of (19) is constrained optimization via Lagrangian duality. Consider

$$\min_{x \in \mathbb{R}^d} \left\{ f(x) + \sum_{i=1}^{n} r_i(x) \right\} \quad \text{s.t.} \quad h_j(x) \leq 0 \quad j = 1, \ldots, p.$$

As in (16), $f$ is a loss function and the $r_i$ may represent regularizers and ("simple") constraints; in addition, there are $p$ functional constraints on the model parameters $x$. Introducing Lagrange multipliers $\gamma \in \mathbb{R}^p$, the problem can be written as

$$\min_{x \in \mathbb{R}^d} \max_{\gamma \in \mathbb{R}^p_+} \left\{ f(x) + \sum_{i=1}^{n} r_i(x) + \sum_{j=1}^{p} \gamma_j h_j(x) \right\}.$$

Under appropriate convexity conditions and constraint-qualifications, this reduces to the following inclusion in the form of (1):

$$0 \in \begin{bmatrix} \nabla f(x) + \sum_{j=1}^{p} \gamma_j \nabla h_j(x) \\ -h(x) \end{bmatrix} + \sum_{i=1}^{n} \left( \partial r_i(x^*) \times \{0\} \right)$$

where $h(x) = [h_1(x), h_2(x), \ldots, h_p(x)]^\top$. For certain choices of $h$, such as linear or quadratic functions, the first term above is monotone and (locally) Lipschitz continuous (Alacaoglu et al., 2021).

**Bilinear Games with Many Constraints**   Finally, consider the bilinear saddlepoint problem subject to multiple constraints:

$$\min_{x \in \mathbb{R}^d} \max_{y \in \mathbb{R}^d} x^\top D y \quad \text{s.t.} \quad x \in \mathcal{C}^1_j \quad j = 1, \ldots, n_1,$$

$$y \in \mathcal{C}^2_j \quad j = 1, \ldots, n_2.$$

Under some regularity conditions, this problem reduces to the inclusion

$$0 \in \begin{bmatrix} Dy^* \\ -D^\top x^* \end{bmatrix} + \sum_{j=1}^{\max\{n_1,n_2\}} \left( N_{\mathcal{C}_j^1}(x^*) \times N_{\mathcal{C}_j^2}(y^*) \right),$$

where we introduce additional "dummy" sets $\mathcal{C}_j^1 = \mathbb{R}^d$ or $\mathcal{C}_j^2 = \mathbb{R}^d$ when $n_1 \neq n_2$. The first term is linear and skew symmetric, and therefore can easily be shown to be Lipschitz continuous and monotone. If all the constraint sets are closed and convex, then the rest of the terms are maximal monotone, then the problem is of the form (1), meaning that projective splitting may be applied, possibly using a stochastic oracle for the first term.

## B    ADDITIONAL RELATED WORK

The preprint by Bot et al. (2019) develops a stochastic version of Tseng's method under the requirement that the noise variance converges to $0$. In ML, this could be achieved with the use of perpetually increasing batch sizes, a strategy that is impractical in many scenarios. The stochastic version of FRB proposed by Van Dung & Vu (2021) has more practical noise requirements, but has stronger assumptions on the problem which are rarely satisfied in ML applications: either uniform/strong monotonicity or a bounded domain. The papers by Yurtsever et al. (2016) and Pedregosa et al. (2019) consider stochastic variants of three-operator splitting, but require $B$ in (1) to be cocoercive, essentially restricting them to optimization problems.

There are several alternatives to the (stochastic) extragradient method that reduce the number of gradient evaluations per iteration from two to one (Hsieh et al., 2019; Malitsky & Tam, 2020; Gidel et al., 2019). However, these methods have more stringent stepsize limits, making it unclear *a priori* whether they will outperform two-step methods.

DSEG is a stochastic version of EG (Hsieh et al., 2020). The primary innovation of DSEG is using different stepsizes for the extrapolation and update steps, thereby resolving some of the convergence issues affecting stochastic EG. As noted earlier, DSEG is the special case of our SPS method in which $n = 0$, that is, no regularizers/constraints are present in the underlying game. The analysis in (Hsieh et al., 2020) also did not consider the fixed stepsize choice given in Theorem 2.

In the context of GANs, several methods have been developed based on a variational inequality/monotone inclusion approach (Gidel et al., 2019; Daskalakis et al., 2018; Hsieh et al., 2019; 2020; Böhm et al., 2020). Many of these papers point out that variational inequalities provide a principled framework for studying the GAN training problem and correcting some of the flaws in the standard method GDA.

## C    PROOF OF THEOREM 1

### C.1    STOCHASTIC QUASI-FEJER MONOTONICITY

The key to the analysis is showing that the algorithm satisfies *Stochastic Quasi-Fejer Monotonicity* (Combettes & Pesquet, 2015).

**Lemma 2** ((Combettes & Pesquet, 2015), Proposition 2.3). *Suppose $p^k$ is a sequence of $\mathbb{R}^d$-valued random variables defined on a probability space $(\Omega, \mathcal{F}, P)$. Let $\mathcal{F}_k = \sigma(p^1, \ldots, p^k)$. Let $F$ be a nonempty, closed subset of $\mathbb{R}^d$. Suppose that, for every $p \in F$, there exists $\chi^k(p) \geq 0, \eta^k(p) \geq 0, \nu^k(p) \geq 0$ such that $\sum_{k=1}^{\infty} \chi^k(p) < \infty$, $\sum_{k=1}^{\infty} \eta^k(p) < \infty$ and*

$$(\forall k \in \mathbb{N}) \quad \mathbb{E}[\|p^{k+1} - p\|^2 | \mathcal{F}_k] \leq (1 + \chi^k(p))\|p^k - p\|^2 - \nu^k(p) + \eta^k(p).$$

*Then the following hold:*

1. *$(\forall p \in F): \quad \sum_{k=1}^{\infty} \nu^k(p) < \infty$ a.s.*

2. *$p^k$ is bounded a.s.*

3. *There exists $\tilde{\Omega}$ such that $P[\tilde{\Omega}] = 1$ and $\{\|p^k(\omega) - p\|\}$ converges for every $\omega \in \tilde{\Omega}$ and $p \in F$.*

## C.2 Important Recursion for SPS

The following lemma summarizes the key recursion satisfied by Algorithm 1, to which we will apply Lemma 2. Recall that $L$ is the Lipschitz constant of $B$.

**Lemma 3.** *For Algorithm 1, suppose* (9)–(11) *hold and*

$$\rho_k \leq \overline{\rho} < 1/L. \tag{20}$$

*Let*

$$T_k \doteq \frac{\tau}{\overline{\rho}} \sum_{i=1}^{n} \|y_i^k - w_i^k\|^2 + \frac{1}{\overline{\rho}\tau} \sum_{i=1}^{n} \|z^k - x_i^k\|^2 + 2(1 - \overline{\rho}L)\|B(z^k) - w_{n+1}^k\|^2$$

*then for all $p^* \in \mathcal{S}$, with probability one*

$$\mathbb{E}[\|p^{k+1} - p^*\|^2 | \mathcal{F}_k] \leq (1 + C_1\alpha_k^2 + C_3\alpha_k\rho_k^2)\|p^k - p^*\|^2 - \alpha_k\rho_k T_k + C_2\alpha_k^2 + C_4\alpha_k\rho_k^2 \tag{21}$$

*where $C_1, \ldots, C_4$ are nonegative constants defined in* (33)*,* (34)*,* (48)*, and* (49) *below, respectively.*

Note that $T_k$ is a scaled version of the approximation residual $G_k$ defined in (14).

We proceed to first prove Lemma 3 and then exploit the implications of Lemma 2. Referring to (10) and (11), let $N \doteq \max_{j \in 1..4} N_j$. To simplify the constants, we will use $N$ in place of $N_j$ for the noise variance bounds given in (10)-(11).

## C.3 Upper Bounding the Gradient

Throughout the analysis, we fix some $p^* = (z^*, w_1^* \ldots, w_{n+1}^*) \in \mathcal{S}$. All statements are with probability one (almost surely), but for brevity we will omit this unless it needs to be emphasized.

In this section, we derive appropriate upper bounds for $\|\nabla\varphi_k\|^2$ to use in (13). We begin with $\nabla_z\varphi_k$:

$$\|\nabla_z\varphi_k\|^2 = \left\|\sum_{i=1}^{n+1} y_i^k\right\|^2 \leq 2\|y_{n+1}^k\|^2 + 2\left\|\sum_{i=1}^{n} y_i^k\right\|^2 = 2\|B(x_{n+1}^k) + e^k\|^2 + 2\left\|\sum_{i=1}^{n} y_i^k\right\|^2$$

$$\leq 4\|B(x_{n+1}^k)\|^2 + 2\left\|\sum_{i=1}^{n} y_i^k\right\|^2 + 4\|e^k\|^2.$$

Now next take expectations with respect to $\mathcal{F}_k$ and $\mathcal{E}_k$, and use the bound on the variance of the noise in (11), obtaining

$$\mathbb{E}\left[\|\nabla_z\varphi_k\|^2 | \mathcal{F}_k, \mathcal{E}_k\right] \leq \mathbb{E}\left[4\|B(x_{n+1}^k)\|^2 + 2\left\|\sum_{i=1}^{n} y_i^k\right\|^2 + 4\|e^k\|^2 \,\Big|\, \mathcal{F}_k, \mathcal{E}_k\right]$$

$$\leq 4(N+1)\|B(x_{n+1}^k)\|^2 + 2\left\|\sum_{i=1}^{n} y_i^k\right\|^2 + 4N,$$

where we have used that $y_i^k$ is $\mathcal{F}_k$-measurable for $i \in 1..n$. Thus, taking expectations over $\mathcal{E}_k$ conditioned on $\mathcal{F}_k$ yields

$$\mathbb{E}\left[\|\nabla_z\varphi_k\|^2 | \mathcal{F}_k\right] \leq 4(N+1)\mathbb{E}[\|B(x_{n+1}^k)\|^2 | \mathcal{F}_k] + 2\left\|\sum_{i=1}^{n} y_i^k\right\|^2 + 4N. \tag{22}$$

We will now bound the two terms on the right side of (22).

### C.3.1 First Term in (22)

First, note that

$$\|B(z^k)\|^2 = \|B(z^k) - B(z^*) + B(z^*)\|^2$$

$$\leq 2\|B(z^k) - B(z^*)\|^2 + 2\|B(z^*)\|^2$$

$$\leq 2L^2\|z^k - z^*\|^2 + 2\|B(z^*)\|^2$$

$$\leq 2L^2\|p^k - p^*\|^2 + 2\|B(z^*)\|^2. \tag{23}$$

Now, returning to the first term on the right of (22), we have

$$
\begin{aligned}
\|B(x_{n+1}^k)\|^2 &= \|B(z^k) + B(x_{n+1}^k) - B(z^k)\|^2 \\
&\leq 2\|B(z^k)\|^2 + 2\|B(x_{n+1}^k) - B(z^k)\|^2 \\
&\leq 2\|B(z^k)\|^2 + 2L^2\|x_{n+1}^k - z^k\|^2 \\
&\leq 4L^2\|p^k - p^*\|^2 + 4\|B(z^*)\|^2 + 2L^2\|x_{n+1}^k - z^k\|^2
\end{aligned}
\tag{24}
$$

where we have used (23) to obtain (24).

For the third term in (24), we have from the calculation on line 7 of the algorithm that

$$
x_{n+1}^k - z^k = -\rho_k(r^k - w_{n+1}^k) = -\rho_k(B(z^k) + \epsilon^k - w_{n+1}^k),
$$

and therefore

$$
\begin{aligned}
\|x_{n+1}^k - z^k\|^2 &= \rho_k^2 \|B(z^k) + \epsilon^k - w_{n+1}^k\|^2 \\
&\leq \bar{\rho}^2 \|B(z^k) + \epsilon^k - w_{n+1}^k\|^2 \\
&\leq 3\bar{\rho}^2 (\|B(z^k)\|^2 + \|\epsilon^k\|^2 + \|w_{n+1}^k\|^2).
\end{aligned}
$$

We next take expectations conditioned on $\mathcal{F}_k$ and use the noise variance bound (10) to obtain

$$
\begin{aligned}
\mathbb{E}\big[\|x_{n+1}^k - z^k\|^2 \mid \mathcal{F}_k\big] &\leq \mathbb{E}\big[3\bar{\rho}^2(\|B(z^k)\|^2 + \|\epsilon^k\|^2 + \|w_{n+1}^k\|^2) \mid \mathcal{F}_k\big] \\
&\leq 3\bar{\rho}^2 \big((N+1)\|B(z^k)\|^2 + \|w_{n+1}^k\|^2 + N\big).
\end{aligned}
$$

Therefore

$$
\begin{aligned}
\mathbb{E}\big[\|x_{n+1}^k - z^k\|^2 \mid \mathcal{F}_k\big] &\leq 6\bar{\rho}^2\big((N+1)\|B(z^k)\|^2 + \|w_{n+1}^k - w_{n+1}^*\|^2 + \|w_{n+1}^*\|^2\big) + 3\bar{\rho}^2 N \\
&= 6\bar{\rho}^2 \Big(2(N+1)L^2\|p^k - p^*\|^2 + 2(N+1)\|B(z^*)\|^2 \\
&\qquad\qquad + \|w_{n+1}^k - w_{n+1}^*\|^2 + \|B(z^*)\|^2\Big) + 3\bar{\rho}^2 N \\
&\leq 6\bar{\rho}^2\big(2(N+1)L^2\|p^k - p^*\|^2 + \|w_{n+1}^k - w_{n+1}^*\|^2\big) \\
&\qquad + 18\bar{\rho}^2(N+1)\|B(z^*)\|^2 + 3\bar{\rho}^2 N \\
&\leq 18\bar{\rho}^2(N+1)\big((L^2+1)\|p^k - p^*\|^2 + \|B(z^*)\|^2\big) + 3\bar{\rho}^2 N
\end{aligned}
\tag{25}
$$

where in the equality uses (23) and $w_{n+1}^* = B(z^*)$. Combining (24) and (25), we arrive at

$$
\begin{aligned}
\mathbb{E}\left[\left.\|B(x_{n+1}^k)\|^2\right| \mathcal{F}_k\right] &\leq 4L^2\big[1 + 9\bar{\rho}^2(L^2+1)(N+1)\big]\|p^k - p^*\|^2 \\
&\qquad + 4\big(1 + 9\bar{\rho}^2 L^2(N+1)\big)\|B(z^*)\|^2 + 6\bar{\rho}^2 L^2 N.
\end{aligned}
\tag{26}
$$

### C.3.2  SECOND TERM IN (22)

For $i \in 1..n$, line 5 of the algorithm may be rearranged into $y_i^k = \tau^{-1}(z^k - x_i^k) + w_i^k$, so

$$
\begin{aligned}
\left\|\sum_{i=1}^n y_i^k\right\|^2 &= \left\|\sum_{i=1}^n (\tau^{-1}(z^k - x_i^k) + w_i^k)\right\|^2 \\
&\leq 2\left\|\tau^{-1}\sum_{i=1}^n (z^k - x_i^k)\right\|^2 + 2\left\|\sum_{i=1}^n w_i^k\right\|^2 \\
&\leq 2n\tau^{-2}\sum_{i=1}^n \|z^k - x_i^k\|^2 + 2\left\|\sum_{i=1}^n w_i^k\right\|^2 \\
&\leq 4n^2\tau^{-2}\|z^k - z^*\|^2 + 4n\tau^{-2}\sum_{i=1}^n \|z^* - x_i^k\|^2 + 4n\sum_{i=1}^n \|w_i^k - w_i^*\|^2 + 4\left\|\sum_{i=1}^n w_i^*\right\|^2 \\
&\leq 4n^2(\tau^{-2}+1)\|p^k - p^*\|^2 + 4n\tau^{-2}\sum_{i=1}^n \|z^* - x_i^k\|^2 + 4\left\|\sum_{i=1}^n w_i^*\right\|^2.
\end{aligned}
\tag{27}
$$

By the definition of the solution set $\mathcal{S}$ in (5), $w_i^* \in A_i(z^*)$, so $z^* + \tau w_i^* \in (I + \tau A_i)(z^*)$, and since the resolvent is single-valued (Bauschke & Combettes, 2017, Cor. 23.9) we therefore obtain

$$z^* = (I + \tau A_i)^{-1}(I + \tau A_i)(z^*) = J_{\tau A_i}(z^* + \tau w_i^*).$$

From lines 3 and 4 of the algorithm, we also have $x_i^k = J_{\tau A_i}(z^k + \tau w_i^k)$ for $i \in 1..n$. Thus, using the nonexpansiveness of the resolvent (Bauschke & Combettes, 2017, Def. 4.1 and Cor. 23.9), we have

$$
\begin{aligned}
\sum_{i=1}^{n} \|z^* - x_i^k\|^2 &= \sum_{i=1}^{n} \left\| J_{\tau A_i}(z^k + \tau w_i^k) - J_{\tau A_i}(z^* + \tau w_i^*) \right\|^2 \\
&\leq \sum_{i=1}^{n} \|z^k + \tau w_i^k - z^* - \tau w_i^*\|^2 \\
&= \sum_{i=1}^{n} \|z^k - z^* + \tau(w_i^k - w_i^*)\|^2 \\
&\leq 2n\|z^k - z^*\|^2 + 2\tau^2 \sum_{i=1}^{n} \|w_i^k - w_i^*\|^2 \\
&\leq 2(n + \tau^2)\|p^k - p^*\|^2.
\end{aligned}
\tag{28}
$$

Combining (27) and (28) yields

$$
\left\| \sum_{i=1}^{n} y_i^k \right\|^2 \leq 12n^2\tau^{-2}(n + \tau^2)\|p^k - p^*\|^2 + 4\left\| \sum_{i=1}^{n} w_i^* \right\|^2.
\tag{29}
$$

Combining (26) and (29) with (22) yields

$$
\begin{aligned}
\mathbb{E}\big[\|\nabla_z \varphi_k\|^2 \,|\, \mathcal{F}_k\big] \leq\ & 24\left[(1 + 9\overline{\rho}^2)(L^2 + 1)^2(N + 1)^2 + n^2\tau^{-2}(n + \tau^2)\right]\|p^k - p^*\|^2 \\
& + 16(N + 1)\big(1 + 9\overline{\rho}^2 L^2(N + 1)\big)\|B(z^*)\|^2 + 8\left\| \sum_{i=1}^{n} w_i^* \right\|^2 \\
& + 24\overline{\rho}^2 L^2(N + 1)N + 4N.
\end{aligned}
\tag{30}
$$

### C.3.3 DUAL GRADIENT NORM

Considering that $\nabla \varphi_k$ is taken with respect to the subspace $\mathcal{P}$, the gradients with respect to the dual variables are — see for example Eckstein & Svaiter (2009) — for each $i \in 1..(n + 1)$,

$$
\begin{aligned}
\|\nabla_{w_i} \varphi_k\|^2 = \left\| x_i^k - \frac{1}{n+1}\sum_{j=1}^{n+1} x_j^k \right\|^2 &= \left\| \frac{1}{n+1}\sum_{j=1}^{n+1}(x_i^k - x_j^k) \right\|^2 \\
&\leq \sum_{j=1}^{n+1} \|x_i^k - x_j^k\|^2 \\
&\leq 2\sum_{j=1}^{n+1} \left(\|x_i^k - z^k\|^2 + \|z^k - x_j^k\|^2\right)
\end{aligned}
$$

Summing this inequality for $i \in 1..(n + 1)$ and collecting terms yields

$$
\sum_{i=1}^{n+1} \|\nabla_{w_i} \varphi_k\|^2 \leq 4(n + 1)\sum_{i=1}^{n+1} \|x_i^k - z^k\|^2,
$$

so taking expectations conditioned on $\mathcal{F}_k$ produces

$$\sum_{i=1}^{n+1} \mathbb{E}[\|\nabla_{w_i}\varphi_k\|^2 \mid \mathcal{F}_k] \leq 4(n+1)\sum_{i=1}^{n+1}\mathbb{E}[\|x_i^k - z^k\|^2 \mid \mathcal{F}_k]$$

$$\leq 4(n+1)\mathbb{E}[\|x_{n+1}^k - z^k\|^2 \mid \mathcal{F}_k] + 4(n+1)\sum_{i=1}^{n}\mathbb{E}[\|x_i^k - z^k\|^2 \mid \mathcal{F}_k]$$

$$\leq 4(n+1)\mathbb{E}[\|x_{n+1}^k - z^k\|^2 \mid \mathcal{F}_k]$$
$$+ 8(n+1)\sum_{i=1}^{n}\mathbb{E}[\|x_i^k - z^*\|^2 \mid \mathcal{F}_k] + 8(n+1)^2\|z^k - z^*\|^2$$

$$\leq 4(n+1)\mathbb{E}[\|x_{n+1}^k - z^k\|^2|\mathcal{F}_k]$$
$$+ 8(n+1)\sum_{i=1}^{n}\mathbb{E}[\|x_i^k - z^*\|^2|\mathcal{F}_k] + 8(n+1)^2\|p^k - p^*\|^2$$

$$\leq 8(n+1)\big[3n + 2\tau^2 + 1 + 9\bar{\rho}^2(L^2+1)(N+1)\big]\|p^k - p^*\|^2$$
$$+ 72\bar{\rho}^2(n+1)(N+1)\|B(z^*)\|^2 + 12\bar{\rho}^2(n+1)N, \tag{31}$$

where the final inequality employs (25) and (28).

All told, using (30) and (31) and simplifying the constants, one obtains

$$\mathbb{E}[\|\nabla\varphi_k\|^2 \mid \mathcal{F}_k] = \mathbb{E}[\|\nabla_z\varphi_k\|^2 \mid \mathcal{F}_k] + \sum_{i=1}^{n+1}\mathbb{E}[\|\nabla_{w_i}\varphi_k\|^2|\mathcal{F}_k]$$

$$\leq C_1\|p^k - p^*\|^2 + C_2, \tag{32}$$

where

$$C_1 = 24(1 + 10\bar{\rho}^2)(n+1)(L^2+1)^2(N+1)^2$$
$$+ 8(n+1)\left(2\tau^2 + 6(n+1) + 1 + 3(n+1)^2\tau^{-2}\right) \tag{33}$$

and

$$C_2 = 16(N+1)\left[1 + 4\bar{\rho}^2(n+1) + 9\bar{\rho}^2 L^2(N+1)\right]\|B(z^*)\|^2 + 8\|\sum_{i=1}^{n}w_i^*\|^2$$
$$+ 12\bar{\rho}^2 N(2L^2(N+1) + n + 1) + 4N. \tag{34}$$

### C.4 Lower Bound for $\varphi_k$-gap

Recalling (13), that is,

$$\|p^{k+1} - p^*\|^2 = \|p^k - p^*\|^2 - 2\alpha_k(\varphi_k(p^k) - \varphi_k(p^*)) + \alpha_k^2\|\nabla\varphi_k\|^2.$$

We may use the gradient bound from (32) to obtain

$$\mathbb{E}[\|p^{k+1} - p^*\|^2 \mid \mathcal{F}_k] \leq (1 + C_1\alpha_k^2)\|p^k - p^*\|^2 - 2\alpha_k\mathbb{E}[\varphi_k(p^k) - \varphi_k(p^*) \mid \mathcal{F}_k] + C_2\alpha_k^2. \tag{35}$$

We now focus on finding a lower bound for the term $\mathbb{E}[\varphi_k(p^k) - \varphi_k(p^*) \mid \mathcal{F}_k]$, which we call the "$\varphi_k$-gap". Recall that for $p = (z, w_1, \ldots, w_{n+1})$,

$$\varphi_k(p) = \sum_{i=1}^{n+1}\langle z - x_i^k, y_i^k - w_i\rangle.$$

For each $i \in 1..(n+1)$, define $\varphi_{i,k}(p) \doteq \langle z - x_i^k, y_i^k - w_i\rangle$. We will call $\mathbb{E}[\varphi_{i,k}(p^k) - \varphi_{i,k}(p^*) \mid \mathcal{F}_k]$ the "$\varphi_{i,k}$-gap". Note that $\varphi_k(p) = \sum_{i=1}^{n+1}\varphi_{i,k}(p)$.

### C.5 Lower Bound for $\varphi_{i,k}$-gap over $i \in 1..n$

For $i \in 1..n$, we have from line 5 of the algorithm that

$$z^k - x_i^k = \tau(y_i^k - w_i^k).$$

Since $\varphi_{i,k}(p^k) = \langle z^k - x_i^k, y_i^k - w_i^k \rangle$, one may conclude that for $i \in 1..n$,

$$\varphi_{i,k}(p^k) = \frac{\tau}{2}\|y_i^k - w_i^k\|^2 + \frac{1}{2\tau}\|z^k - x_i^k\|^2.$$

On the other hand, for $p^* \in \mathcal{S}$ and $i \in 1..n$, one also has

$$-\varphi_{i,k}(p^*) = \langle z^* - x_i^k, w_i^* - y_i^k \rangle \geq 0 \tag{36}$$

by the monotonicity of $A_i$. Therefore, for $i \in 1..n$, it holds that

$$\varphi_{i,k}(p^k) - \varphi_{i,k}(p^*) \geq \frac{\tau}{2}\|y_i^k - w_i^k\|^2 + \frac{1}{2\tau}\|z^k - x_i^k\|^2,$$

and taking expectations conditioned on $\mathcal{F}_k$ leads to

$$\mathbb{E}[\varphi_{i,k}(p^k) - \varphi_{i,k}(p^*) \mid \mathcal{F}_k] \geq \frac{\tau}{2}\|y_i^k - w_i^k\|^2 + \frac{1}{2\tau}\|z^k - x_i^k\|^2 \tag{37}$$

where we have used that $x_i^k$ and $y_i^k$ are both $\mathcal{F}_k$-measurable for $i \in 1..n$.

### C.6 Lower Bound for $\varphi_{n+1,k}$-gap

From lines 6-7 of the algorithm, we have

$$z^k - x_{n+1}^k = \rho_k(B(z^k) - w_{n+1}^k + \epsilon^k).$$

Therefore,

$$
\begin{aligned}
\varphi_{n+1,k}(p^k) &= \langle z^k - x_{n+1}^k, y_{n+1}^k - w_{n+1}^k \rangle \tag{38}\\
&= \langle z^k - x_{n+1}^k, B(z^k) - w_{n+1}^k \rangle + \langle z^k - x_{n+1}^k, y_{n+1}^k - B(z^k) \rangle \\
&= \rho_k\langle B(z^k) - w_{n+1}^k + \epsilon^k, B(z^k) - w_{n+1}^k \rangle + \langle z^k - x_{n+1}^k, y_{n+1}^k - B(z^k) \rangle \\
&= \rho_k\|B(z^k) - w_{n+1}^k\|^2 + \langle z^k - x_{n+1}^k, y_{n+1}^k - B(z^k) \rangle + \rho_k\langle \epsilon^k, B(z^k) - w_{n+1}^k \rangle \\
&\overset{(a)}{=} \rho_k\|B(z^k) - w_{n+1}^k\|^2 + \langle z^k - x_{n+1}^k, B(x_{n+1}^k) - B(z^k) \rangle + \langle z^k - x_{n+1}^k, \epsilon^k \rangle \\
&\qquad + \rho_k\langle \epsilon^k, B(z^k) - w_{n+1}^k \rangle \\
&\geq \rho_k\|B(z^k) - w_{n+1}^k\|^2 - L\|z^k - x_{n+1}^k\|^2 + \langle z^k - x_{n+1}^k, \epsilon^k \rangle \\
&\qquad + \rho_k\langle \epsilon^k, B(z^k) - w_{n+1}^k \rangle \\
&= \rho_k\|B(z^k) - w_{n+1}^k\|^2 - L\|\rho_k(B(z^k) - w_{n+1}^k + \epsilon^k)\|^2 + \langle z^k - x_{n+1}^k, \epsilon^k \rangle \\
&\qquad + \rho_k\langle \epsilon^k, B(z^k) - w_{n+1}^k \rangle \\
&= \rho_k\|B(z^k) - w_{n+1}^k\|^2 - \rho_k^2 L\|B(z^k) - w_{n+1}^k + \epsilon^k\|^2 + \langle z^k - x_{n+1}^k, \epsilon^k \rangle \\
&\qquad + \rho_k\langle \epsilon^k, B(z^k) - w_{n+1}^k \rangle \\
&= \rho_k(1 - \rho_k L)\|B(z^k) - w_{n+1}^k\|^2 - \rho_k^2 L\|\epsilon^k\|^2 + \langle z^k - x_{n+1}^k, \epsilon^k \rangle \\
&\qquad + \rho_k(1 - 2\rho_k L)\langle \epsilon^k, B(z^k) - w_{n+1}^k \rangle, \tag{39}
\end{aligned}
$$

where equality (a) uses line 8 of the algorithm and the inequality employs the Cauchy-Schwartz inequality followed by Lipschitz continuity of $B$.

On the other hand,

$$
\begin{aligned}
-\varphi_{n+1,k}(p^*) &= \langle z^* - x_{n+1}^k, w_{n+1}^* - y_{n+1}^k \rangle \\
&= \langle z^* - x_{n+1}^k, B(z^*) - B(x_i^k) \rangle + \langle x_{n+1}^k - z^*, \epsilon^k \rangle \\
&\geq \langle x_{n+1}^k - z^*, \epsilon^k \rangle, \tag{40}
\end{aligned}
$$

where the second equality uses line 8 of the algorithm and the inequality follows from the monotonicity of $B$.

Combining (39) and (40) yields

$$
\begin{aligned}
\varphi_{n+1,k}(p^k) - \varphi_{n+1,k}(p^*) &\geq \rho_k(1 - \rho_k L)\|B(z^k) - w_{n+1}^k\|^2 + \rho_k(1 - 2\rho_k L)\langle \epsilon^k, B(z^k) - w_{n+1}^k\rangle \\
&\quad + \langle z^k - x_{n+1}^k, e^k\rangle + \langle x_{n+1}^k - z^*, e^k\rangle - \rho_k^2 L\|\epsilon^k\|^2 \\
&= \rho_k(1 - \rho_k L)\|B(z^k) - w_{n+1}^k\|^2 - \rho_k^2 L\|\epsilon^k\|^2 \\
&\quad + \rho_k(1 - 2\rho_k L)\langle \epsilon^k, B(z^k) - w_{n+1}^k\rangle + \langle z^k - z^*, e^k\rangle.
\end{aligned}
\tag{41}
$$

Now, if we take expectations conditioned on $\mathcal{F}_k$ and use (9), we obtain

$$
\mathbb{E}\big[\langle z^k - z^*, e^k\rangle \,\big|\, \mathcal{F}_k\big] = \langle z^k - z^*, \mathbb{E}[e^k \,|\, \mathcal{F}_k]\rangle = 0.
\tag{42}
$$

Similarly, (9) also yields

$$
\mathbb{E}\big[\langle \epsilon^k, B(z^k) - w_{n+1}^k\rangle \,\big|\, \mathcal{F}_k\big] = \big\langle \mathbb{E}[\epsilon^k|\mathcal{F}_k], B(z^k) - w_{n+1}^k\big\rangle = 0.
\tag{43}
$$

Thus, using (42) and (43) and taking expectations of (41) yields

$$
\begin{aligned}
\mathbb{E}[\varphi_{n+1,k}(p^k) - \varphi_{n+1,k}(p^*)\,|\,\mathcal{F}_k] &\geq \rho_k(1 - \rho_k L)\|B(z^k) - w_{n+1}^k\|^2 - \rho_k^2 L\mathbb{E}[\|\epsilon^k\|^2|\mathcal{F}_k] \\
&\geq \rho_k(1 - \overline{\rho}L)\|B(z^k) - w_{n+1}^k\|^2 - \rho_k^2 NL(1 + \|B(z^k)\|^2),
\end{aligned}
\tag{44}
$$

where in the second inequality we used (12) and the noise variance bound (10). Recall from (12) that $1 - \overline{\rho}L > 0$.

Next, we remark that

$$
\begin{aligned}
\|B(z^k)\|^2 &= \|B(z^k) - B(z^*) + B(z^*)\|^2 \\
&\leq 2L^2\|z^k - z^*\|^2 + 2\|B(z^*)\|^2 \leq 2L^2\|p^k - p^*\|^2 + 2\|B(z^*)\|^2.
\end{aligned}
$$

Substituting this inequality into (44) yields

$$
\begin{aligned}
\mathbb{E}[\varphi_{n+1,k}(p^k) - \varphi_{n+1,k}(p^*)|\mathcal{F}_k] &\geq \rho_k(1 - \overline{\rho}L)\|B(z^k) - w_{n+1}^k\|^2 \\
&\quad - 2\rho_k^2 NL^3\|p^k - p^*\|^2 - \rho_k^2 NL(1 + 2\|B(z^*)\|^2).
\end{aligned}
\tag{45}
$$

**Finalizing the lower bound on the $\varphi_k$-gap**   Summing (37) over $i \in 1..n$ and using (45) yields

$$
\begin{aligned}
\mathbb{E}[\varphi_k(p^k) - \varphi_k(p^*)|\mathcal{F}_k] &= \sum_{i=1}^{n+1} \mathbb{E}[\varphi_{i,k}(p^k) - \varphi_{i,k}(p^*)|\mathcal{F}_k] \\
&\geq \frac{\tau}{2}\sum_{i=1}^{n}\|y_i^k - w_i^k\|^2 + \frac{1}{2\tau}\sum_{i=1}^{n}\|z^k - x_i^k\|^2 \\
&\quad + \rho_k(1 - \overline{\rho}L)\|B(z^k) - w_{n+1}^k\|^2 - 2\rho_k^2 NL^3\|p^k - p^*\|^2 \\
&\quad - \rho_k^2 NL(1 + 2\|B(z^*)\|^2).
\end{aligned}
\tag{46}
$$

## C.7   Establishing Stochastic Quasi-Fejér Monotonicity

Returning to (35),

$$
\mathbb{E}[\|p^{k+1} - p^*\|^2 \,|\, \mathcal{F}_k] \leq (1 + C_1\alpha_k^2)\|p^k - p^*\|^2 - 2\alpha_k\mathbb{E}[\varphi_k(p^k) - \varphi_k(p^*) \,|\, \mathcal{F}_k] + C_2\alpha_k^2,
$$

we may now substitute (46) for the expectation on the right-hand side. First, define

$$
T_k \doteq \frac{\tau}{\overline{\rho}}\sum_{i=1}^{n}\|y_i^k - w_i^k\|^2 + \frac{1}{\overline{\rho}\tau}\sum_{i=1}^{n}\|z^k - x_i^k\|^2 + 2(1 - \overline{\rho}L)\|B(z^k) - w_{n+1}^k\|^2,
$$

after which we may use (46) in (35) to yield

$$
\mathbb{E}[\|p^{k+1} - p^*\|^2 \,|\, \mathcal{F}_k] \leq (1 + C_1\alpha_k^2 + C_3\alpha_k\rho_k^2)\|p^k - p^*\|^2 - \alpha_k\rho_k T_k + C_2\alpha_k^2 + C_4\alpha_k\rho_k^2
\tag{47}
$$

where $C_1$ and $C_2$ are defined as before in (33) and (34) and

$$
C_3 = 4NL^3
\tag{48}
$$

$$
C_4 = 2NL(1 + 2\|B(z^*)\|^2).
\tag{49}
$$

This completes the proof of Lemma 3.

## C.8 A CONVERGENCE LEMMA

Before establishing almost-sure convergence, we need the following lemma to derive convergence of the iterates from convergence of $T_k$ defined above. Note that a more elaborate result would be needed in an infinite-dimensional setting.

**Lemma 4.** *For deterministic sequences $z^k \in \mathbb{R}^{(n+1)d}, \{(w_i^k)_{i=1}^{n+1}\} \in \mathcal{P}$, and $\{(x_i^k, y_i^k)_{i=1}^{n+1}\} \in \mathbb{R}^{2(n+1)d}$, suppose that $y_i^k \in A_i(x_i^k)$ for $i \in 1..n$, $\sum_{i=1}^{n+1} w_i^k = 0$,*

$$\xi_1 \sum_{i=1}^n \|y_i^k - w_i^k\|^2 + \xi_2 \sum_{i=1}^n \|z^k - x_i^k\|^2 + \xi_3 \|B(z^k) - w_{n+1}^k\|^2 \to 0 \tag{50}$$

*for scalars $\xi_1, \xi_2, \xi_3 > 0$, and $p^k \doteq (z^k, w_1^k, \dots, w_{n+1}^k) \to \hat{p} \doteq (\hat{z}, \hat{w}_1, \dots, \hat{w}_{n+1})$. Then $\hat{p} \in \mathcal{S}$.*

*Proof.* Fix any $i \in \{1, \dots, n\}$. Since $\|y_i^k - w_i^k\| \to 0$ by (50) and $w_i^k \to \hat{w}_i$, we also have $y_i^k \to \hat{w}_i$. Similarly, (50) also implies that $\|z^k - x_i^k\| \to 0$, so from $z^k \to \hat{z}$ we also have $x_i^k \to \hat{z}$. Since $y_i^k \in A_i(x_i^k)$ and $(x_i^k, y_i^k) \to (\hat{z}, \hat{w}_i)$, (Bauschke & Combettes, 2017, Prop. 20.37) implies $\hat{w}_i \in A_i(\hat{z})$. Since $i$ was arbitrary, the preceding conclusions hold for $i \in 1..n$.

Now, (50) also implies that $\|B(z^k) - w_{n+1}^k\| \to 0$. Therefore, since $w_{n+1}^k \to \hat{w}_{n+1}$, we also have $B(z^k) \to \hat{w}_{n+1}$. Much as before, since $(z^k, B(z^k)) \to (\hat{z}, \hat{w}_{n+1})$, we may apply (Bauschke & Combettes, 2017, Prop. 20.37) to conclude that that $\hat{w}_{n+1} = B(\hat{z})$.

Since the linear subspace $\mathcal{P}$ defined in (6) must be closed, the limit $(\hat{z}, \hat{w}_1, \dots, \hat{w}_{n+1})$ of $\{(z^k, w_1^k, \dots, w_{n+1}^k)\} \subset \mathcal{P}$ must be in $\mathcal{P}$, hence $\sum_{i=1}^{n+1} \hat{w}_i = 0$.

Thus, the point $\hat{p} = (\hat{z}, \hat{w}_1, \dots, \hat{w}_{n+1})$ satisfies $\hat{w}_i \in A_i(\hat{z})$ for $i \in 1..n$, $\hat{w}_{n+1} = B(\hat{z})$, and $\sum_{i=1}^{n+1} \hat{w}_i = 0$. These are the three conditions defining membership in $\mathcal{S}$ from (5), so $\hat{p} \in \mathcal{S}$. □

## C.9 FINISHING THE PROOF OF THEOREM 1

Given $\sum_k \alpha_k^2 < \infty$, and $\sum \alpha_k \rho_k^2 < \infty$, (47) satisfies the conditions of *Stochastic Quasi-Fejer Monotonicity* as given in Lemma 2. By applying Lemma 2, we conclude that there exist $\Omega_1, \Omega_2, \Omega_3$ such that $P[\Omega_i] = 1$ for $i = 1, 2, 3$ and

1. for all $v \in \Omega_1$

$$\sum_{k=1}^{\infty} \alpha_k \rho_k T_k(v) < \infty, \tag{51}$$

2. for all $v \in \Omega_2$, and $p^* \in \mathcal{S}$, $\|p^k(v) - p^*\|$ converges to a finite nonnegative random-variable,
3. for all $v \in \Omega_3$, $p^k(v)$ remains bounded.

Since $\sum_{k=1}^{\infty} \alpha_k \rho_k = \infty$, (51) implies that for all $v \in \Omega_1$ there exists a subsequence $q_k(v)$ such that

$$T_{q_k(v)} \to 0. \tag{52}$$

Let $\Omega' = \Omega_1 \cap \Omega_2 \cap \Omega_3$ and note that $P[\Omega'] = 1$. Choose $v \in \Omega'$. Since $p^k(v)$ remains bounded, so does $p^{q_k(v)}(v)$ for $q_k(v)$ defined above in (52). Thus there exists a subsequence $r_k(v) \subseteq q_k(v)$ and $\hat{p}(v) \in \mathbb{R}^{(n+2)d}$ such that $p^{r_k(v)}(v) \to \hat{p}(v)$. But since $T_{q_k(v)} \to 0$, it also follows that $T_{r_k(v)} \to 0$, that is,

$$\frac{\tau}{\rho} \sum_{i=1}^n \|y_i^{r_k(v)}(v) - w_i^{r_k(v)}(v)\|^2 + \frac{1}{\rho\tau} \sum_{i=1}^n \|z^{r_k(v)}(v) - x_i^{r_k(v)}(v)\|^2$$
$$+ 2(1 - \bar{\rho}L)\|B(z^{r_k(v)}(v)) - w_{n+1}^{r_k(v)}(v)\|^2 \to 0.$$

We then have from Lemma 4 that $\hat{p}(v) \in \mathcal{S}$.

Since $p^{r_k(v)}(v) \to \hat{p}(v)$, it follows that $\|p^{r_k(v)}(v) - \hat{p}(v)\| \to 0$. But since $\hat{p}(v) \in \mathcal{S}$, $\|p^k(v) - \hat{p}(v)\|$ converges by point 2 above. Thus

$$\lim_{k \to \infty} \|p^k(v) - \hat{p}(v)\| = \lim_{k \to \infty} \|p^{r_k(v)}(v) - \hat{p}(v)\| = 0.$$

Therefore $p^k(v) \to \hat{p}(v) \in \mathcal{S}$. Thus there exists $\hat{p} \in \mathcal{S}$ such that $p^k \to \hat{p}$ a.s., which completes the proof of Theorem 1.

### C.10 TWO ADDITIONAL RESULTS

In this section, we prove two additional useful results about SPS. First, that $x_i^k \to \hat{z}$ (a.s.) for $i = 1, \ldots, n$. Second, that $G_k \to 0$ (a.s.).

Note that

$$x_i^k = J_{\tau A_i}(z^k + \tau w_i^k)$$

and since $z^k$ and $w_i^k$ convergence a.s., so does $x_i^k$. Consider the subsequence $q_k(v)$ such that (52) holds. Then

$$z^{q_k(v)} - x_i^{q_k(v)} \to 0$$

thus

$$x_i^{q_k(v)} \to \hat{z}.$$

Since $x_i^k$ converges to some limit (a.s.), that limit must be $\hat{z}$.

Recall that

$$G_k \doteq \sum_{i=1}^n \|y_i^k - w_i^k\|^2 + \sum_{i=1}^n \|z^k - x_i^k\|^2 + \|B(z^k) - w_{n+1}^k\|^2.$$

We have shown that $z^k$ and $x_i^k$ share the same limit for $i = 1, \ldots, n$ (a.s.). Therefore $z^k - x_i^k \to 0$ (a.s.). Since

$$y_i^k - w_i^k = \tau^{-1}(z^k - x_i^k),$$

it follows that $y_i^k - w_i^k \to 0$ (a.s.) for $i = 1, \ldots, n$. Therefore

$$G_k \to \|B(\hat{z}) - \hat{w}_{n+1}\|^2.$$

But since $(z, \hat{w}_1, \ldots, \hat{w}_{n+1}) \in \mathcal{S}$, $\hat{w}_{n+1} = B(\hat{z})$ implying that $G_k \to 0$ (a.s.).

## D   PROOF OF LEMMA 1

If $G_k = 0$, then

$$\forall i = 1, \ldots, n : \quad y_i^k = w_i^k \text{ and } z^k = x_i^k. \tag{53}$$

Since $y_i^k \in A_i(x_i^k)$ for $i = 1, \ldots, n$, (53) implies that that

$$\forall i \in 1..n : \quad w_i^k \in A_i(z^k). \tag{54}$$

Furthermore $G_k = 0$ also implies that $w_{n+1}^k = B(z^k)$. Finally, since $\sum_{i=1}^{n+1} w_i^k = 0$, we have that

$$(z^k, w_1^k, \ldots, w_{n+1}^k) \in \mathcal{S}.$$

Conversely, suppose $(z^k, w_1^k, \ldots, w_{n+1}^k) \in \mathcal{S}$. The definition of $\mathcal{S}$ implies that $B(z^k) = w_{n+1}^k$ and furthermore that $w_i^k \in A_i(z^k)$ for $i \in 1..n$. For any $i \in 1..n$, considering line 3 of Algorithm 1, we may write $t_i^k = z^k + \tau w_i^k \in (I + \tau A_i)(z^k)$, implying $z^k \in (I + \tau A_i)^{-1}(t_i^k)$. But since the resolvent $J_{\tau A_i} = (I + \tau A_i)^{-1}$ is single-valued (Bauschke & Combettes, 2017, Prop. 23.8), we must have $z^k = (I + \tau A_i)^{-1}(t_i^k)$. Thus, by line 4, we have $x_i^k = z^k$. We may also derive from line 5 that

$$y_i^k = \tau^{-1}(t_i^k - x_i^k) = \tau^{-1}(z^k + \tau w_i^k - z^k) = w_i^k.$$

Thus, since $x_i^k = z^k$ and $y_i^k = w_i^k$ for $i = 1, \ldots, n$ and $w_{n+1}^k = B(z^k)$, we have that $G_k = 0$.

## E    PROOF OF THEOREM 2

In addition to the proof, we provide a more detailed statement of the theorem:

**Theorem 3.** *Fix the total iterations $K \geq 1$ of Algorithm 1 and set*

$$\forall k = 1, \ldots, K: \qquad\qquad \rho_k = \rho \doteq \min\left\{ K^{-1/4}, \frac{1}{2L} \right\} \qquad (55)$$

$$\forall k = 1, \ldots, K: \qquad\qquad \alpha_k = \alpha \doteq C_f \rho^2 \qquad (56)$$

*for some $C_f > 0$. Suppose (9)-(11) hold. Then for any $p^* \in \mathcal{S}$,*

$$\frac{1}{K} \sum_{j=1}^{K} \mathbb{E}[G_j] \leq \frac{8L^3 \exp\left(C_f(C_1 + C_3)\right)}{C_f \min\{\tau, \tau^{-1}\} K} \left( \|p^1 - p^*\|^2 + \frac{C_f C_2 + C_4}{C_f C_1 + C_3} \right) \quad \text{for } K < (2L)^4 \quad (57)$$

$$\frac{1}{K} \sum_{j=1}^{K} \mathbb{E}[G_j] \leq \frac{\exp\left(C_f(C_1 + C_3)\right)}{C_f \min\{\tau, \tau^{-1}\} K^{1/4}} \left( \|p^1 - p^*\|^2 + \frac{C_f C_2 + C_4}{C_f C_1 + C_3} \right) \qquad \text{for } K \geq (2L)^4. \quad (58)$$

*where $G_k$ is the approximation residual defined in (14), and $C_1, C_2, C_3, C_4$ are the nonegative constants defined in (33), (34), (48), and (49), respectively. Therefore,*

$$\frac{1}{K} \sum_{j=1}^{K} \mathbb{E}[G_j] = \mathcal{O}(K^{-1/4}).$$

*Proof.* Fix $\alpha_k = \alpha$ and $\rho_k = \rho$, where $\alpha$ and $\rho$ are the respective right-hand sides of (55)-(56). Lemma 3 implies that (21) so long as (9)-(11) hold and the stepsize $\rho$ satisfies $\rho < L^{-1}$. Since

$$\rho = \min\left\{ K^{-1/4}, \frac{1}{2L} \right\} \leq \frac{1}{2L},$$

we conclude that (21) applies.

Rewriting (21) with $\alpha_k = \alpha$ and $\rho_k = \rho$, we have

$$\mathbb{E}[\|p^{k+1} - p^*\|^2 \mid \mathcal{F}_k] \leq (1 + C_1 \alpha^2 + C_3 \alpha \rho^2) \|p^k - p^*\|^2 - \alpha \rho T_k + C_2 \alpha^2 + C_4 \alpha \rho^2.$$

Therefore, taking expectations over $\mathcal{F}_k$, we have

$$\mathbb{E}\|p^{k+1} - p^*\|^2 \leq (1 + C_1 \alpha^2 + C_3 \alpha \rho^2) \mathbb{E}\|p^k - p^*\|^2 - \alpha \rho \mathbb{E} T_k + C_2 \alpha^2 + C_4 \alpha \rho^2. \qquad (59)$$

Recall that

$$T_k \doteq \frac{\tau}{\rho} \sum_{i=1}^{n} \|y_i^k - w_i^k\|^2 + \frac{1}{\rho \tau} \sum_{i=1}^{n} \|z^k - x_i^k\|^2 + 2(1 - \overline{\rho}L) \|B(z^k) - w_{n+1}^k\|^2,$$

where for the first two terms we have simply set $\rho = \overline{\rho}$ because the stepsize is constant. However, for the final term, we will still use an upper bound, $\overline{\rho}$, on $\rho$. In the current setting, we know that $\rho \leq (1/2)L^{-1}$ and therefore we may set $\overline{\rho} = (1/2)L^{-1}$. Thus $1 - \overline{\rho}L = 1/2$, leading to

$$\rho \mathbb{E} T_k = \tau \sum_{i=1}^{n} \mathbb{E}\|y_i^k - w_i^k\|^2 + \tau^{-1} \sum_{i=1}^{n} \mathbb{E}\|z^k - x_i^k\|^2 + \rho \mathbb{E}\|B(z^k) - w_{n+1}^k\|^2.$$

Let

$$U_k \doteq \mathbb{E}\|B(z^k) - w_{n+1}^k\|^2 \qquad W_k \doteq \tau \sum_{i=1}^{n} \mathbb{E}\|y_i^k - w_i^k\|^2 + \tau^{-1} \sum_{i=1}^{n} \mathbb{E}\|z^k - x_i^k\|^2,$$

so that

$$\rho \mathbb{E} T_k = \rho U_k + W_k,$$

and also let

$$V_k \doteq \mathbb{E}\|p^k - p^*\|^2.$$

Using these definitions in (59) we write

$$V_{k+1} \le (1 + C_1\alpha^2 + C_3\alpha\rho^2)V_k - \alpha\rho U_k - \alpha W_k + C_2\alpha^2 + C_4\alpha\rho^2.$$

Therefore,

$$V_{k+1} + \alpha\rho U_k + \alpha W_k \le (1 + C_1\alpha^2 + C_3\alpha\rho^2)V_k + C_2\alpha^2 + C_4\alpha\rho^2$$

$$\iff V_{k+1} + \alpha\rho\sum_{j=1}^{k} U_j + \alpha\sum_{j=1}^{k} W_j \le (1 + C_1\alpha^2 + C_3\alpha\rho^2)V_k + \alpha\rho\sum_{j=1}^{k-1} U_j + \alpha\sum_{j=1}^{k-1} W_j$$

$$+ C_2\alpha^2 + C_4\alpha\rho^2$$

$$\le (1 + C_1\alpha^2 + C_3\alpha\rho^2)\left[V_k + \alpha\rho\sum_{j=1}^{k-1} U_j + \alpha\sum_{j=1}^{k-1} W_j\right]$$

$$+ C_2\alpha^2 + C_4\alpha\rho^2,$$

where we have used that $U_k, W_k \ge 0$. Letting

$$R_k = V_k + \alpha\rho\sum_{j=1}^{k-1} U_j + \alpha\sum_{j=1}^{k-1} W_j,$$

we then have

$$R_{k+1} \le (1 + C_1\alpha^2 + C_3\alpha\rho^2)R_k + C_2\alpha^2 + C_4\alpha\rho^2,$$

which implies

$$R_{k+1} \le (1 + C_1\alpha^2 + C_3\alpha\rho^2)^k R_1 + (C_2\alpha^2 + C_4\alpha\rho^2)\sum_{j=1}^{k}(1 + C_1\alpha^2 + C_3\alpha\rho^2)^{k-j}.$$

Now,

$$\sum_{j=1}^{k}(1 + C_1\alpha^2 + C_3\alpha\rho^2)^{k-j} = \sum_{j=0}^{k-1}(1 + C_1\alpha^2 + C_3\alpha\rho^2)^j$$

$$= \frac{(1 + C_1\alpha^2 + C_3\alpha\rho^2)^k - 1}{(1 + C_1\alpha^2 + C_3\alpha\rho^2) - 1}$$

$$= \frac{(1 + C_1\alpha^2 + C_3\alpha\rho^2)^k - 1}{C_1\alpha^2 + C_3\alpha\rho^2}$$

$$\le \frac{(1 + C_1\alpha^2 + C_3\alpha\rho^2)^k}{C_1\alpha^2 + C_3\alpha\rho^2}.$$

Therefore,

$$R_{k+1} \le (1 + C_1\alpha^2 + C_3\alpha\rho^2)^k\left(R_1 + \frac{C_2\alpha^2 + C_4\alpha\rho^2}{C_1\alpha^2 + C_3\alpha\rho^2}\right).$$

Fix the number of iterations $K \ge 1$. Now

$$\rho = \min\left\{K^{-1/4}, \frac{1}{2L}\right\} \le \frac{1}{K^{1/4}} \le 1.$$

Therefore,

$$\alpha\rho\sum_{j=1}^{K}(U_j + W_j) \le \alpha\rho\sum_{j=1}^{K} U_j + \alpha\sum_{j=1}^{K} W_j$$

$$\le R_{K+1}$$

$$\le (1 + C_1\alpha^2 + C_3\alpha\rho^2)^K\left(R_1 + \frac{C_2\alpha^2 + C_4\alpha\rho^2}{C_1\alpha^2 + C_3\alpha\rho^2}\right).$$

Dividing through by $\alpha\rho K$, we obtain

$$\frac{1}{K}\sum_{j=1}^{K}(U_j + W_j) \le \frac{(1 + C_1\alpha^2 + C_3\alpha\rho^2)^K}{\alpha\rho K}\left(R_1 + \frac{C_2\alpha^2 + C_4\alpha\rho^2}{C_1\alpha^2 + C_3\alpha\rho^2}\right), \tag{60}$$

and since $\alpha = C_f\rho^2$, we also have

$$\frac{C_2\alpha^2 + C_4\alpha\rho^2}{C_1\alpha^2 + C_3\alpha\rho^2} = \frac{C_fC_2 + C_4}{C_fC_1 + C_3}.$$

Furthermore,

$$\rho \le K^{-\frac{1}{4}} \implies \alpha \le C_f K^{-\frac{1}{2}}.$$

Substituting these into (60) yields

$$\frac{1}{K}\sum_{j=1}^{K}(U_j + W_j) \le \frac{\left(1 + \frac{C_f(C_fC_1 + C_3)}{K}\right)^K}{\alpha\rho K}\left(R_1 + \frac{C_fC_2 + C_4}{C_fC_1 + C_3}\right)$$

$$\le \frac{\exp(C_f(C_fC_1 + C_3))}{\alpha\rho K}\left(R_1 + \frac{C_fC_2 + C_4}{C_fC_1 + C_3}\right), \tag{61}$$

where we have used that for any $t \ge 0$, $1 + t/K \le e^{t/K}$, so therefore $(1 + t/K)^K \le e^t$.

The worst-case rates in terms of $K$ occur when $\rho = K^{-1/4}$ and $\alpha = C_f K^{-1/2}$. This is the case when $K \ge (2L)^4$. Substituting these into the denominator yields, for $K \ge (2L)^4$, that

$$\frac{1}{K}\sum_{j=1}^{K}(U_j + W_j) \le \frac{\exp(C_f(C_fC_1 + C_3))}{C_f K^{1/4}}\left(R_1 + \frac{C_fC_2 + C_4}{C_fC_1 + C_3}\right).$$

Thus, since $G_k \le \max\{\tau, \tau^{-1}\}(U_k + W_k)$, we obtain

$$\frac{1}{K}\sum_{j=1}^{K}\mathbb{E}[G_j] \le \frac{\max\{\tau, \tau^{-1}\}\exp\left(C_f(C_fC_1 + C_3)\right)}{C_f K^{1/4}}\left(\|p^1 - p^*\|^2 + \frac{C_fC_2 + C_4}{C_fC_1 + C_3}\right),$$

which is (58).

When $K < (2L)^4$, (57) can similarly be obtained by substituting $\rho = (2L)^{-1}$ and $\alpha = C_f(2L)^{-2}$ into (61). $\qquad\square$

## F  APPROXIMATION RESIDUALS

In this section we derive the approximation residual used to assess the performance of the algorithms in the numerical experiments. This residual relies on the following product-space reformulation of (1).

### F.1  PRODUCT-SPACE REFORMULATION AND RESIDUAL PRINCIPLE

Recall (1), the monotone inclusion we are solving:

$$\text{Find } z \in \mathbb{R}^d : 0 \in \sum_{i=1}^{n} A_i(z) + B(z).$$

In this section we demonstrate a "product-space" reformulation of (1) which allows us to rewrite it in a standard form involving just two operators, one maximal monotone and the other monotone and Lipschitz. This approach was pioneered in (Briceño-Arias & Combettes, 2011; Combettes & Pesquet, 2012). Along with allowing for a simple definition of an approximation residual as a measure of approximation error in solving (1), it allows one to apply operator splitting methods originally formulated for two operators to problems such as (1) for any finite $n$.

Observe that solving (1) is equivalent to

$$\text{Find } (w_1, \ldots, w_n, z) \in \mathbb{R}^{(n+1)d} : \quad w_i \in A_i(z), \quad i \in 1..n$$

$$0 \in \sum_{i=1}^{n} w_i + B(z).$$

This formulation resembles that of the extended solution set $\mathcal{S}$ used in projective spitting, as given in (5), except that it combines the final two conditions in the definition of $\mathcal{S}$, and thus does not need the final dual variable $w_{n+1}$. From the definition of the inverse of an operator, the above formulation is equivalent to

$$\text{Find } (w_1, \ldots, w_n, z) \in \mathbb{R}^{(n+1)d} : \quad 0 \in A_i^{-1}(w_i) - z, \quad i \in 1..n$$

$$0 \in \sum_{i=1}^{n} w_i + B(z).$$

These conditions are in turn equivalent to finding $(w_1, \ldots, w_n, z) \in \mathbb{R}^{(n+1)d}$ such that

$$0 \in \mathscr{A}(w_1, \ldots, w_n, z) + \mathscr{B}(w_1, \ldots, w_n, z), \tag{62}$$

where $\mathscr{A}$ is the set-valued map

$$\mathscr{A}(w_1, \ldots, w_n, z) \mapsto A_1^{-1}(w_1) \times A_2^{-1}(w_2) \times \ldots \times A_n^{-1}(w_n) \times \{0\} \tag{63}$$

and $\mathscr{B}$ is the single-valued operator

$$\mathscr{B}(w_1, \ldots, w_n, z) \mapsto \begin{bmatrix} 0 & \cdots & 0 & -I \\ \vdots & \ddots & \vdots & \vdots \\ 0 & \cdots & 0 & -I \\ I & \cdots & I & 0 \end{bmatrix} \begin{bmatrix} w_1 \\ \vdots \\ w_n \\ z \end{bmatrix} + \begin{bmatrix} 0 \\ \vdots \\ 0 \\ B(z) \end{bmatrix}. \tag{64}$$

It is easily established that $\mathscr{B}$ is maximal monotone and Lipschitz continuous, while $\mathscr{A}$ is maximal monotone. Letting $\mathscr{T} \doteq \mathscr{A} + \mathscr{B}$, it follows from (Bauschke & Combettes, 2017, Proposition 20.23) that $\mathscr{T}$ is maximal monotone. Thus, we have reformulated (1) as the monotone inclusion $0 \in \mathscr{T}(q)$ for $q$ in the product space $\mathbb{R}^{(n+1)d}$. A vector $z \in \mathbb{R}^d$ solves (1) if and only if there exists $(w_1, \ldots, w_n) \in \mathbb{R}^{nd}$ such that $0 \in \mathscr{T}(q)$, where $q = (w_1, \ldots, w_n, z)$.

For any pair $(q, v)$ such that $v \in \mathscr{T}(q)$, $\|v\|^2$ represents an *approximation residual* for $q$ in the sense that $v = 0$ implies $q$ is a solution to (62). One may take $\|v\|^2$ as a measure of the error of $q$ as an approximate solution to (62), and it can only be 0 if $q$ is a solution. Given two approximate solutions $q_1$ and $q_2$ with certificates $v_1 \in T(q_1)$ and $v_2 \in \mathscr{T}(q_2)$, we will treat $q_1$ as a "better" approximate solution than $q_2$ if $\|v_1\|^2 < \|v_2\|^2$. Doing so is somewhat analogous to the practice, common in optimization, of using the gradient $\|\nabla f(x)\|^2$ as a measure of quality of an approximate minimizer of some differentiable function $f$. However, note that since $\mathscr{T}(q_1)$ is a set, there may exist elements of $\mathscr{T}(q_1)$ with smaller norm than $v_1$. Thus any given certificate $v_1$ only corresponds to an upper bound on $\text{dist}^2(0, \mathscr{T}(q_1))$.

### F.2 APPROXIMATION RESIDUAL FOR PROJECTIVE SPLITTING

In SPS (Algorithm 1), for $i \in 1..n$, the pairs $(x_i^k, y_i^k)$ are chosen so that $y_i^k \in A_i(x_i^k)$. This can be seen from the definition of the resolvent. Thus $x_i^k \in A_i^{-1}(y_i^k)$. Observe that

$$v^k \doteq \begin{bmatrix} x_1^k - z^k \\ \vdots \\ x_n^k - z^k \\ B(z^k) + \sum_{i=1}^{n} y_i^k \end{bmatrix} \in \mathscr{T}(y_1^k, \ldots, y_n^k, z^k). \tag{65}$$

The approximation residual for SPS is thus

$$R_k \doteq \|v^k\|^2 = \sum_{i=1}^{n} \|z^k - x_i^k\|^2 + \left\| B(z^k) + \sum_{i=1}^{n} y_i^k \right\|^2 \tag{66}$$

which is an approximation residual for $(y_1^k, \ldots, y_n^k, z^k)$ in the sense defined above. We may relate $R_k$ to the approximation residual $G_k$ for SPS from Section 5 as follows:

$$
\begin{aligned}
R_k &= \sum_{i=1}^n \|z^k - x_i^k\|^2 + \left\|B(z^k) + \sum_{i=1}^n y_i^k\right\|^2 \\
&= \sum_{i=1}^n \|z^k - x_i^k\|^2 + \left\|B(z^k) + \sum_{i=1}^n y_i^k - \sum_{i=1}^{n+1} w_i^k\right\|^2 \\
&\leq \sum_{i=1}^n \|z^k - x_i^k\|^2 + 2\|B(z^k) - w_{n+1}^k\|^2 + 2\left\|\sum_{i=1}^n (y_i^k - w_i^k)\right\|^2 \\
&\leq \sum_{i=1}^n \|z^k - x_i^k\|^2 + 2\|B(z^k) - w_{n+1}^k\|^2 + 2n \sum_{i=1}^n \left\|y_i^k - w_i^k\right\|^2 \\
&\leq 2nG_k
\end{aligned}
$$

where in the second equality we have used the fact that $\sum_{i=1}^{n+1} w_i^k = 0$. Thus, $R_k$ has the same convergence rate as $G_k$ given in Theorem 2.

Note that while the certificate given in (65) focuses on the primal iterate $z^k$, it may be changed to focus on any $x_i^k$ for $i = 1, \ldots, n$, by using

$$
v_i^k \doteq \begin{bmatrix} x_1^k - x_i^k \\ \vdots \\ x_n^k - x_i^k \\ B(x_i^k) + \sum_{i=1}^n y_i^k \end{bmatrix} \in \mathscr{T}(y_1^k, \ldots, y_n^k, x_i^k).
$$

The approximation residual $\|v_i^k\|^2$ may also be shown to have the same rate as $G_k$ by following similar derivations to those above for $R_k$.

### F.3 TSENG'S METHOD

Tseng's method (Tseng, 2000) can be applied to (62), resulting in the following recursion with iterates $q^k, \bar{q}^k \in \mathbb{R}^{(n+1)d}$:

$$
\bar{q}^k = J_{\alpha \mathscr{A}}(q^k - \alpha \mathscr{B}(q^k)) \tag{67}
$$

$$
q^{k+1} = \bar{q}^k + \alpha\big(\mathscr{B}(q^k) - \mathscr{B}(\bar{q}^k)\big), \tag{68}
$$

where $\mathscr{A}$ and $\mathscr{B}$ are defined in (63) and (64). The resolvent of $\mathscr{A}$ may be readily computed from the resolvents of the $A_i$ using Moreau's identity (Bauschke & Combettes, 2017, Proposition 23.20).

Analogous to SPS, Tseng's method has an approximation residual, which in this case is an element of $\mathscr{T}(\bar{q}^k)$. In particular, using the general properties of resolvent operators as applied to $J_{\alpha \mathscr{A}}$, we have

$$
\frac{1}{\alpha}(q^k - \bar{q}^k) - \mathscr{B}(q^k) \in \mathscr{A}(\bar{q}^k).
$$

Also, rearranging (68) produces

$$
\frac{1}{\alpha}(\bar{q}^k - q^{k+1}) + \mathscr{B}(q^k) = \mathscr{B}(\bar{q}^k).
$$

Adding these two relations produces

$$
\frac{1}{\alpha}(q^k - q^{k+1}) \in \mathscr{A}(\bar{q}^k) + \mathscr{B}(\bar{q}^k) = \mathscr{T}(\bar{q}^k)
$$

Therefore,

$$
R_k^{\text{Tseng}} \doteq \frac{1}{\alpha^2}\|q^k - q^{k+1}\|^2
$$

represents a measure of the approximation error for Tseng's method equivalent to $R_k$ defined in (66) for SPS.

### F.4 FRB

The forward-reflected-backward method (FRB) (Malitsky & Tam, 2020) is another method that may be applied to the splitting $\mathscr{T} = \mathscr{A} + \mathscr{B}$ for $\mathscr{A}$ and $\mathscr{B}$ as defined in (63) and (64). Doing so yields recursion

$$q^{k+1} = J_{\alpha\mathscr{A}}\Big(q^k - \alpha\big(2\mathscr{B}(q^k) - \mathscr{B}(q^{k-1})\big)\Big).$$

Following similar arguments to those for Tseng's method, it can be shown that

$$v^k_{\text{FRB}} \doteq \frac{1}{\alpha}\left(q^{k-1} - q^k\right) + \mathscr{B}(q^k) + \mathscr{B}(q^{k-2}) - 2\mathscr{B}(q^{k-1}) \in \mathscr{T}(q^k).$$

Thus, FRB admits the following approximation residual equivalent to $R_k$ for SPS:

$$R^{\text{FRB}}_k \doteq \|v^k_{\text{FRB}}\|^2.$$

Finally, we remark that the stepsizes used in both the Tseng and FRB methods can be chosen via a linesearch procedure that we do not detail here.

### F.5 STOCHASTIC TSENG METHOD

The stochastic version of Tseng's method of (Böhm et al., 2020) (S-Tseng) may be applied to the inclusion $0 \in \mathscr{A}(q) + \mathscr{B}(q)$, since the operator $\mathscr{A}$ may be written as a subdifferential. However, unlike the deterministic Tseng method, it does not produce a valid residual. Note also that S-Tseng outputs an ergodic sequence $q^k_{\text{erg}}$. To construct a residual for the ergodic sequence, we compute a deterministic step of Tseng's method according to (67)-(68), starting at $q^k_{\text{erg}}$. That is, letting

$$\bar{q}^k = J_{\alpha\mathscr{A}}(q^k_{\text{erg}} - \mathscr{B}(q^k_{\text{erg}}))$$
$$q^{k+1} = \bar{q}^k + \alpha(\mathscr{B}(q^k_{\text{erg}}) - \mathscr{B}(\bar{q}^k)),$$

we can then compute essentially the same residual as in Section F.3,

$$R^{\text{S-Tseng}}_k \doteq \frac{1}{\alpha^2}\|q^k_{\text{erg}} - q^{k+1}\|^2.$$

To construct the stochastic oracle for S-Tseng, we assumed $B(z) = \frac{1}{m}\sum_{i=1}^m B_i(z)$. Then we used

$$\tilde{\mathscr{B}}(w_1, \ldots, w_n, z) \mapsto \begin{bmatrix} 0 & \cdots & 0 & -I \\ \vdots & \ddots & \vdots & \vdots \\ 0 & \cdots & 0 & -I \\ I & \cdots & I & 0 \end{bmatrix} \begin{bmatrix} w_1 \\ \vdots \\ w_n \\ z \end{bmatrix} + \begin{bmatrix} 0 \\ \vdots \\ 0 \\ \frac{1}{|\mathbf{B}|}\sum_{j\in\mathbf{B}} B_j(z) \end{bmatrix}. \quad (69)$$

for some minibatch $\mathbf{B} \in \{1, \ldots, m\}$.

### F.6 VARIANCE-REDUCED FRB

The FRB-VR method of Alacaoglu et al. (2021) can also be applied to $0 \in \mathscr{A}(q) + \mathscr{B}(q)$, using the same stochastic oracle $\tilde{\mathscr{B}}$ defined in (69). if we let the iterates of FRB-VR be $(q^k, p^k)$, then line 4 of Algorithm 1 of Alacaoglu et al. (2021) can be written as

$$\hat{q}^k = q^k - \tau(\mathscr{B}(p^k) + \tilde{\mathscr{B}}(q^k) - \tilde{\mathscr{B}}(p^k)) \quad (70)$$
$$q^{k+1} = J_{\tau\mathscr{A}}(\hat{q}^k). \quad (71)$$

Once again, the method does not directly produce a residual, but one can be developed from the algorithm definition as follows: (71) yields $\tau^{-1}(\hat{q}^k - q^{k+1}) \in \mathscr{A}(q^{k+1})$ and hence

$$\tau^{-1}(\hat{q}^k - q^{k+1}) + \mathscr{B}(q^{k+1}) \in (\mathscr{A} + \mathscr{B})(q^{k+1}).$$

Therefore we use the residual

$$R^{\text{FRB-VR}}_k = \|\tau^{-1}(\hat{q}^k - q^{k+1}) + \mathscr{B}(q^{k+1})\|^2.$$

Figure 1 plots $R_k$ for SPS, $R^{\text{Tseng}}_k$ for Tseng's method, $R^{\text{FRB}}_k$ for FRB, $R^{\text{S-Tseng}}_k$ for S-Tseng, and $R^{\text{FRB-VR}}_k$ for FRB-VR.

### F.7    Benefits and Drawbacks of the Product Space Reformulation

The main benefit of the product space reformulation (PSR) is that it allows one to use familiar 2-operator splitting schemes for solving $0 \in \mathscr{A}(q) + \mathscr{B}(q)$ to solve the more complicated recursion (1). However, one drawback of this approach is that the operator $\mathscr{B}$, defined in (64), combines a skew-symmetric consensus matrix with the Lipschitz operator $B$. Treating $\mathscr{B}$ as a single operator necessitates using a single stepsize for both of its constituent operators, but the $B$ component will generally have a much larger Lipschitz constant than the skew part, necessitating a smaller stepsize than is ideal for the skew operator. This difficulty can be countered by using different stepsizes for the primal and dual components, but that strategy introduces additional tuning parameters. In other works, methods based on PSR have exhibited slower convergence than deterministic projective splitting methods (Johnstone & Eckstein, 2021; 2020b). However, in our experiments in Section 7, the performance is comparable.

## G    Variational Inequalities

For a mapping $B : \mathbb{R}^d \to \mathbb{R}^d$ and a closed and convex set $\mathcal{C}$, the variational inequality problem (Harker & Pang, 1990) is to find $z^* \in \mathcal{C}$ such that

$$B(z^*)^\top (z - z^*) \geq 0, \forall z \in \mathcal{C}. \tag{72}$$

Consider the normal cone mapping discussed in Section 2 and defined as

$$N_\mathcal{C}(x) \doteq \{g : g^\top (y - x) \leq 0 \ \forall y \in \mathcal{C}\}$$

It is easily seen that (72) is equivalent to finding $z^*$ such that $-B(z^*) \in N_\mathcal{C}(z^*)$. Hence, if $B$ is monotone, (72) is equivalent to the monotone inclusion

$$0 \in B(z^*) + N_\mathcal{C}(z^*). \tag{73}$$

Thus, monotone variational inequalities are a special case of monotone inclusions with two operators, one of which is single-valued and the other is the normal cone map of the constraint set $\mathcal{C}$. As a consequence, methods for monotone inclusions can be used to solve monotone variational inequality problems. The reverse, however, may not be true. For example, the analysis of the extragradient method (Korpelevich, 1977) relies on the second operator $N_\mathcal{C}$ in (73) being a normal cone, as opposed to a more general monotone operator. We are not aware of any direct extension of the extragradient method's analysis allowing a more general resolvent to be used in place of the projection map corresponding to $N_\mathcal{C}$.

**The Restricted Gap Function**   There is a disadvantage to pursuing convergence rates based on variational inequalities (as in Böhm et al. (2020) and Alacaoglu et al. (2021)) rather than monotone inclusions. Convergence rate analyses for variational inequalities focus on the *gap function*:

$$G_\mathcal{C}(z) \doteq \sup_{z' \in \mathcal{C}} B(z')^\top (z - z'). \tag{74}$$

It can be shown that $G_\mathcal{C}(z) \geq 0$ and $G_\mathcal{C}(z) = 0$ if and only if $z$ solves (72). However, (74) is meaningless for most problems, since unless $\mathcal{C}$ is compact, $G_\mathcal{C}(z)$ is typically equal to $+\infty$ for any nonsolution (Diakonikolas, 2020). Thus researchers instead focus on the *restricted* gap function (Nesterov, 2007)

$$G_{\mathcal{C}_2}(z) \doteq \sup_{z' \in \mathcal{C}_2} B(z')^\top (z - z'). \tag{75}$$

where $\mathcal{C}_2$ is an arbitrary compact set. However, now the results are only meaningful over the set $\mathcal{C}_2$. Thus, $\mathcal{C}_2$ must be chosen large enough so that the iterates of the algorithm remain in the interior of $\mathcal{C}_2$ (Böhm et al., 2020). Further, the convergence rate bound depends on the diameter of $\mathcal{C}_2$. For some algorithms (Mokhtari et al., 2020) a valid set is provided which bounds the iterates. However Böhm et al. (2020) and Alacaoglu et al. (2021) do not provide one, although in principle it could be done so long as the ergodic sequence can be bounded almost-surely. Thus, the convergence rates depending on (75) in Böhm et al. (2020) and Alacaoglu et al. (2021) are somewhat incomplete in that they depend on unknown constants.

In contrast, rates based on the approximation residual in the monotone inclusion setting, including ours given in (57)–(58), completely avoid this pitfall. There is no need to select a compact set containing the algorithm's iterates and the constants in our rates are all explicit or depend on standard quantities such as the initial distance to a solution.

## H  Memory-Saving Technique for SPS

The variables $t_i^k$, $x_i^k$, and $y_i^k$ on lines 3-5 of SPS are stored in variables $t, x$ and $y$. Another two variables $\bar{x}$ and $\bar{y}$ keep track of $\sum_{i=1}^n x_i^k$ and $\sum_{i=1}^n y_i^k$. The dual variables are stored as $w_i$ for $i \in 1..n$ and the primal variable as $z$. Once $x = x_i^k$ is computed, the $i^{\text{th}}$ dual variable $w_i$ can be partially updated as $w_i \leftarrow w_i - \alpha_k x$. Once all the operators have been processed, the update for each dual variable may be completed via $w_i \leftarrow w_i + \alpha_k (n+1)^{-1} \bar{x}$. Also, the primal update is computed as $z \leftarrow z - \alpha_k \bar{y}$. During the calculation loop for the $x_i^k, y_i^k$, the terms in approximation residual $R_k$ may also be accumulated one by one. The total total number of vector elements that must be stored is $(n+7)d$.

## I  Additional Information About the Numerical Experiments

We solve the following convex-concave min-max problem:

$$\min_{\substack{\beta \in \mathbb{R}^d \\ \lambda \in \mathbb{R}}} \max_{\gamma \in \mathbb{R}^m} \left\{ \lambda(\delta - \kappa) + \frac{1}{m} \sum_{i=1}^m \Psi(\langle \hat{x}_i, \beta \rangle) + \frac{1}{m} \sum_{i=1}^m \gamma_i(\hat{y}_i \langle \hat{x}_i, \beta \rangle - \lambda\kappa) + c\|\beta\|_1 \right\} \tag{76}$$
$$\text{s.t.} \quad \|\beta\|_2 \le \lambda/(L_\Psi + 1) \qquad \|\gamma\|_\infty \le 1.$$

This model is identical to that of (Yu et al., 2021, Thm. 4.3) except for the addition of the $\ell_1$ regularization term $c\|\beta\|_1$, where $c \ge 0$ is a given constant. The goal is to learn the model weights $\beta$ from a training dataset of $m$ feature vectors $\hat{x}_i$ and corresponding labels $\hat{y}_i$. Rather than computing the expected loss over the training set, the formulation uses, for each $\beta$, the worst possible distribution within a Wasserstein-metric ball around the empirical distribution of the $\{(\hat{x}_i, \hat{y}_i)\}$, with the parameter $\delta \ge 0$ giving the diameter of the ball and the parameter $\kappa \ge 0$ specifying the relative weighting of features and labels. The variables $\gamma$ and $\lambda$ parameterize the selection of this worst-case distribution in response to the model weights $\beta$. Finally, $\Psi$ is the logistic loss kernel $t \mapsto \log(e^t + e^{-t})$ and $L_\Psi = 1$ is the corresponding Lipschitz constant. In all the experiments, we set $\delta = \kappa = 1$ and $c = 10^{-3}$.

We now show how we converted this problem to the form (1) for our experiments. Let $z$ be a shorthand for $(\lambda, \beta, \gamma)$, and define

$$\mathcal{L}(z) \doteq \lambda(\delta - \kappa) + \frac{1}{m} \sum_{i=1}^m \Psi(\langle \hat{x}_i, \beta \rangle) + \frac{1}{m} \sum_{i=1}^m \gamma_i(\hat{y}_i \langle \hat{x}_i, \beta \rangle - \lambda\kappa).$$

The first-order necessary and sufficient conditions for the convex-concave saddlepoint problem in (76) are

$$0 \in B(z) + A_1(z) + A_2(z) \tag{77}$$

where the vector field $B(z)$ is defined as

$$B(z) \doteq \begin{bmatrix} \nabla_{\lambda,\beta}\mathcal{L}(z) \\ -\nabla_\gamma \mathcal{L}(z) \end{bmatrix}, \tag{78}$$

with

$$\nabla_{\lambda,\beta}\mathcal{L}(z) = \begin{bmatrix} \delta - \kappa(1 + \frac{1}{m}\sum_{i=1}^m \gamma_i) \\ \frac{1}{m}\sum_{i=1}^m \Psi'(\langle \hat{x}_i, \beta \rangle)\hat{x}_i + \frac{1}{m}\sum_{i=1}^m \gamma_i \hat{y}_i \hat{x}_i \end{bmatrix}$$

and

$$\nabla_\gamma \mathcal{L}(z) = \begin{bmatrix} \frac{1}{m}(\hat{y}_1\langle \hat{x}_1, \beta \rangle - \lambda\kappa) \\ \vdots \\ \frac{1}{m}(\hat{y}_m\langle \hat{x}_m, \beta \rangle - \lambda\kappa) \end{bmatrix}.$$

It is readily confirmed that $B$ defined in this manner is Lipschitz. The monotonicity of $B$ follows from its being the generalized gradient of a convex-concave saddle function (Rockafellar, 1970).

For the set-valued operators, $A_1(z)$ corresponds to the constraints and $A_2(z)$ to the nonsmooth $\ell_1$ regularizer, and are defined as

$$A_1(z) \doteq N_{\mathcal{C}_1}(\lambda, \beta) \times N_{\mathcal{C}_2}(\gamma),$$

where

$$\mathcal{C}_1 \doteq \{(\lambda, \beta) : \|\beta\|_2 \leq \lambda/(L_\Psi + 1)\} \quad \text{and} \quad \mathcal{C}_2 \doteq \{\gamma : \|\gamma\|_\infty \leq 1\},$$

and

$$A_2(z) \doteq \{\mathbf{0}_{1\times 1}\} \times c\partial\|\beta\|_1 \times \{\mathbf{0}_{m\times 1}\}.$$

Here, the notation $\mathbf{0}_{p\times 1}$ denotes the $p$-dimensional vector of all zeros. $\mathcal{C}_1$ is a scaled version of the second-order cone, well known to be a closed convex set, while $\mathcal{C}_2$ is the unit ball of the $\ell_\infty$ norm, also closed and convex. Since $A_1$ is a normal cone map of a closed convex set and $A_2$ is the subgradient map of a closed proper convex function (the scaled 1-norm), both of these operators are maximal monotone and problem (77) is a special case of (1) for $n = 2$.

**Stochastic oracle implementation** The operator $B : \mathbb{R}^{m+d+1} \mapsto \mathbb{R}^{m+d+1}$, defined in (78), can be written as

$$B(z) = \frac{1}{m} \sum_{i=1}^m B_i(z)$$

where

$$B_i(z) \doteq \begin{bmatrix} \delta - \kappa(1 + \gamma_i) \\ \Psi'(\langle \hat{x}_i, \beta \rangle)\hat{x}_i + \gamma_i \hat{y}_i \hat{x}_i \\ \mathbf{0}_{(i-1)\times 1} \\ -(\hat{y}_i\langle \hat{x}_i, \beta \rangle - \lambda\kappa) \\ \mathbf{0}_{(m-i)\times 1} \end{bmatrix}.$$

In our SPS experiments, the stochastic oracle for $B$ is simply $\tilde{B}(z) = \frac{1}{|\mathbf{B}|} \sum_{i\in\mathbf{B}} B_i(z)$ for some minibatch $\mathbf{B} \subseteq \{1, \dots, m\}$. We used a batchsize of 100.

**Resolvent computations** The resolvent of $A_1$ is readily constructed from the projection maps of the simple sets $\mathcal{C}_1$ and $\mathcal{C}_2$, while the resolvent $A_2$ involves the proximal operator of the $\ell_1$ norm. Specifically,

$$J_{\rho A_1}(z) = \begin{bmatrix} \text{proj}_{\mathcal{C}_1}(\lambda, \beta) \\ \text{proj}_{\mathcal{C}_2}(\gamma) \end{bmatrix} \quad \text{and} \quad J_{\rho A_2}(z) = \begin{bmatrix} \mathbf{0}_{1\times 1} \\ \text{prox}_{\rho c\|\cdot\|_1}(\beta) \\ \mathbf{0}_{m\times 1} \end{bmatrix}.$$

The constraint $\mathcal{C}_1$ is a scaled second-order cone and $\mathcal{C}_2$ is the $\ell_\infty$ ball, both of which have closed-form projections. The proximal operator of the $\ell_1$ norm is the well-known soft-thresholding operator (Parikh & Boyd, 2013, Section 6.5.2). Therefore all resolvents in the formulation may be computed quickly and accurately.

**SPS stepsize choices** For the stepsize in SPS, we ordinarily require $\rho_k \leq \overline{\rho} < 1/L$ for the global Lipschitz constant $L$ of $B$. However, since the global Lipschitz constant may be pessimistic, better performance can often be achieved by experimenting with larger stepsizes. If divergence is observed, then the stepsize can be decreased. This type of strategy is common for SGD and similar stochastic methods. Thus, for SPS-decay we set $\alpha_k = C_d k^{-0.51}$ and $\rho_k = C_d k^{-0.25}$, and performed a grid search to select the best $C_d$ from $\{0.1, 0.5, 1, 5, 10\}$, arriving at $C_d = 1$ for epsilon and SUSY, and $C_d = 0.5$ for real-sim. For SPS-fixed we used $\rho = K^{-1/4}$ and $\alpha = C_f \rho^2$, and performed a grid search to select $C_f$ over $\{0.1, 0.5, 1, 5, 10\}$, arriving at $C_f = 1$ for epsilon and real-sim, and $C_f = 5$ for SUSY. The total number of iterations for SPS-fixed was chosen as follows: For the epsilon dataset, we used $K = 5000$, for SUSY we used $K = 200$, and for real-sim we used $K = 1000$.

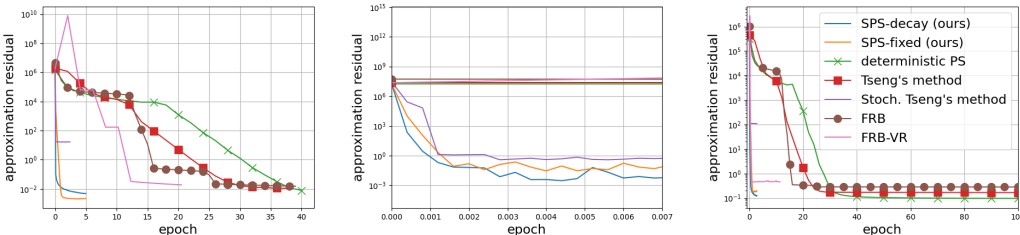

Figure 2: Approximation residual versus epoch for three LIBSVM benchmark datasets. Left: epsilon, middle: SUSY, right: real-sim.

**Parameter choices for the other algorithms**  All methods are initialized at the same random point. For Tseng's method, we used the backtracking linesearch variant with an initial stepsize of $1$, $\theta = 0.8$, and a stepsize reduction factor of $0.7$. For FRB, we used the backtracking linesearch variant with the same settings as for Tseng's method. For deterministic PS, we used a fixed stepsize of $0.9/L$. For the stochastic Tseng's method of Böhm et al. (2020), the stepsize $\alpha_k$ must satisfy: $\sum_{k=1}^{\infty} \alpha_k = \infty$ and $\sum_{k=1}^{\infty} \alpha_k^2 < \infty$. So we set $\alpha_k = Ck^{-d}$ and perform a grid search over $\{C, d\}$ in the range $[10^{-4}, 10] \times [0.51, 1]$, checking $5 \times 5$ values to find the best setting for each of the three problems. The selected values are in Table 1.

| | epsilon | SUSY | real-sim |
|---|---|---|---|
| $C$ | 0.56 | 0.56 | 0.77 |
| $d$ | 0.6 | 0.6 | 0.55 |

Table 1: Parameter Values for S-Tseng

The work of Böhm et al. (2020) also introduced FBFp, a stochastic version of Tseng's method that reuses a previously-computed gradient and therefore only needs one additional gradient calculation per iteration. In our experiments, the performance of the two methods was about the same, so we only report the performance of stoch. Tseng's method.

For variance-reduced FRB, the main parameter is the probability $p$. We hand-tuned $p$, arriving at $p = 0.01$ for all problems. We set the stepsize to its maximum allowed value of

$$\tau = \frac{1 - \sqrt{1-p}}{2L}.$$

**Plots versus Epoch**  Figure 2 plots the performance of each method versus epoch (i.e. data pass). This shows an even more dramatic benefit for the stochastic methods than the plots versus time, since at each iteration the stochastic methods only need to process small amounts of data, whereas deterministic methods must process all of it. We believe these benefits do not fully manifest themselves in the plots versus time due to overheads in each iteration of the stochastic methods, multithreading providing a boost for the deterministic methods, memory access patterns, and other practical considerations.

**Fraction of Nonzero Entries versus Running time**  Figure 3 plots the fraction of nonzero entries in the iterates of each method versus running time. For each method, we used output of $\text{prox}_{c\|\cdot\|_1}$. We observe that our methods produce sparse intermediate iterates for two of the three problems. This is one of the benefits of proximal splitting algorithms in general, including our method. For the other problem, SUSY, no method produces sparse iterates, suggesting that $c$ should be increased if sparse solutions are desired.

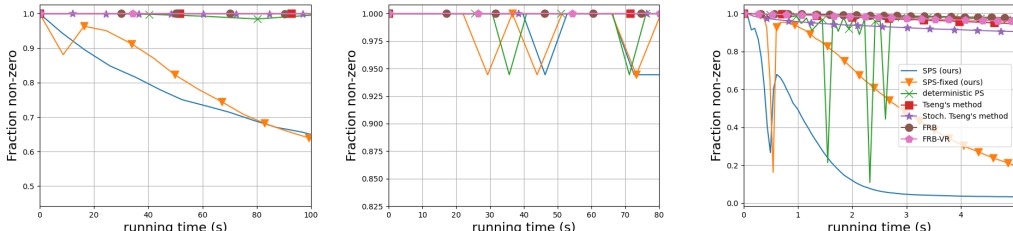

Figure 3: Fraction of nonzero entries versus running time for the three datasets. Left: epsilon, middle: SUSY, right: real-sim.

## J   LOCAL CONVERGENCE ON NON-MONOTONE PROBLEMS

The work by Hsieh et al. (2020) provides a local convergence analysis for DSEG applied to locally monotone problems. Recall that DSEG is equivalent to the special case of SPS for which $n = 0$. While extending this result to the more general setting of SPS is beyond the scope of this manuscript, we next provide a preliminary sketch of how the analysis of Hsieh et al. (2020) might be generalized to our setting. We leave a formal proof to future work.

**Sketch of assumptions and main result**   The first assumption needed is the existence of an isolated solution $p^* = (z^*, w_1^*, \ldots, w_{n+1}^*) \in \mathcal{S}$. We then require that there exists a ball $\mathbb{B}_r(z^*)$, centered at $z^*$, throughout which the operator $B$ is "well-behaved", meaning that it satisfies monotonicity and Lipschitz continuity. In addition, we need each $A_i$, for $i \in 1..n$, to be maximal monotone within this ball. Outside of the ball, the operators do not need to be monotone or Lipschitz.

Following (Hsieh et al., 2020, Assumption 2′), the noise variance assumptions are slightly stronger than in the monotone case. In particular, we require that $\mathbb{E}[\|\epsilon^k\|^q | \mathcal{F}_k] \le N^q$ and $\mathbb{E}[\|e^k\|^q | \mathcal{F}_k] \le N^q$ for some $q > 2$. As before, the noise must be zero-mean. Finally, the stepsize requirements are also slightly stronger than (12), having the added assumption that $\sum_{k=1}^{\infty} \rho_k^q < \infty$.

With these assumptions, the goal is to show that, so long as the initial point $p^1$ is sufficiently close to $p^*$, then with high probability $p^k$ converges to $p^*$.

**Proof strategy**   The initial strategy is to develop the following recursion, satisfied by SPS, that does not (yet) utilize local monotonicity or Lipschitz continuity:

$$\|p^{k+1} - p^*\|^2 \le (1 + c_1 \alpha_k^2)\|p^k - p^*\|^2 - c_2 \alpha_k \rho_k (T_k' + l_k + r_k) - c_3 \alpha_k (r_k' + q_k)$$
$$+ c_1 \alpha_k^2 \big(\|e^k\|^2 + \|\epsilon^k\|^2 + c_4\big) + c_5 \alpha_k q_k' \tag{79}$$

for appropriate constants $c_1 \ldots c_5 \ge 0$. In this inequality, we use

$$T_k' \doteq \frac{\tau}{\rho} \sum_{i=1}^{n} \|y_i^k - w_i^k\|^2 + \frac{1}{\rho \tau} \sum_{i=1}^{n} \|z^k - x_i^k\|^2,$$

$$l_k \doteq \sum_{i=1}^{n} \langle z^* - x_i^k, w_i^* - y_i^k \rangle + \langle z^* - x_{n+1}^k, w_i^* - B(x_{n+1}^k) \rangle,$$

$$r_k \doteq \langle \epsilon^k, B(\tilde{x}^k) - w_{n+1}^k \rangle,$$

$$r_k' \doteq \langle z^k - z^*, e^k \rangle,$$

$$q_k \doteq (\rho_k^{-1} - d/2)\|\tilde{x}^k - z^k\|^2 - \|\tilde{x}^k - z^k\|\|B(\tilde{x}^k) - B(z^k)\|$$

$$q_k' \doteq \rho_k \|\epsilon^k\| \|Bx_{n+1}^k - B\tilde{x}^k\| + \frac{1}{2d}\|B\tilde{x}_{n+1}^k - Bx_{n+1}^k\|^2,$$

where

$$\tilde{x}^k \doteq z^k - \rho_k\big(B(z^k) - w_{n+1}^k\big) \qquad\qquad d \doteq \frac{1 - \bar{\rho}L}{1 + \bar{\rho}/2}, \tag{80}$$

with $L$ being the local Lipschitz constant of $B$ on $\mathbb{B}_r(z^*)$. The iterate $\tilde{x}^k$ is the analog of the iterate $\tilde{X}_{t+1/2}$ used in Hsieh et al. (2020).

The recursion (79) is derived by once again starting from (13) and following the arguments leading to (35), but this time not taking conditional expectations. In particular, the upper bounds on $\|\nabla_z \varphi_k\|^2$ and $\|\nabla_{w_i} \varphi_k\|^2$ contribute the terms $c_1 \alpha_k^2 (\|e^k\|^2 + \|\epsilon^k\|^2 + c_4)$ and $c_1 \alpha_k^2 \|p^k - p^*\|^2$. For $i \in 1..n$, the "$\varphi_{i,k}$-gap" term, $\varphi_{i,k}(p^k) - \varphi_{i,k}(p^*)$, is dealt with in a similar manner to Section C.5, but this time not using monotonicity as in (36). This contributes $T_k'$ and the first term in $l_k$. Finally, as we sketch below, the "$\varphi_{n+1,k}$-gap" term contributes $r_k$, $r_k'$, $q_k$, $q_k'$, and the last term in $l_k$.

For the "$\varphi_{n+1,k}$-gap", that is, $\varphi_{n+1,k}(p^k) - \varphi_{n+1,k}(p^*)$, we have to depart from the analysis in Section C.6 and use an alternative argument involving $\tilde{x}^k$. We now provide some details of this argument: in the following, we use $Bz$ as shorthand for $B(z)$ for any vector $z \in \mathbb{R}^d$. We begin the analysis with

$$\begin{aligned}
\varphi_{n+1,k}(p^k) &= \langle z^k - x_{n+1}^k, y_{n+1}^k - w_{n+1}^k \rangle \\
&= \langle z^k - x_{n+1}^k, Bx_{n+1}^k - w_{n+1}^k \rangle + \underbrace{\langle z^k - x_{n+1}^k, e^k \rangle}_{\text{part of } r_k'}.
\end{aligned} \tag{81}$$

The final term will combine with the term $\langle x_{n+1}^k - z^*, e^k \rangle$ coming from

$$\begin{aligned}
-\varphi_{n+1,k}(p^*) &= \langle z^* - x_{n+1}^k, w_{n+1}^* - y_{n+1}^k \rangle \\
&= \langle z^* - x_{n+1}^k, w_{n+1}^* - Bx_{n+1}^k \rangle + \langle x_{n+1}^k - z^*, e_{n+1}^k \rangle
\end{aligned} \tag{82}$$

to yield $r_k'$ above. Equation (82) also yields the second term in $l_k$. Using that $\tilde{x}^k - x_{n+1}^k = \rho_k \epsilon_k$, we rewrite the first term in (81) as

$$\begin{aligned}
\langle z^k - x_{n+1}^k, Bx_{n+1}^k - w_{n+1}^k \rangle &= \langle z^k - \tilde{x}^k, Bx_{n+1}^k - w_{n+1}^k \rangle + \langle \tilde{x}^k - x_{n+1}^k, Bx_{n+1}^k - w_{n+1}^k \rangle \\
&= \langle z^k - \tilde{x}^k, Bx_{n+1}^k - w_{n+1}^k \rangle + \rho_k \langle \epsilon^k, Bx_{n+1}^k - w_{n+1}^k \rangle \\
&= \langle z^k - \tilde{x}^k, Bx_{n+1}^k - w_{n+1}^k \rangle + \rho_k \langle \epsilon^k, Bx_{n+1}^k - B\tilde{x}^k \rangle \\
&\quad + \rho_k \underbrace{\langle \epsilon^k, B\tilde{x}^k - w_{n+1}^k \rangle}_{r_k}.
\end{aligned} \tag{83}$$

Next, the terms in (83) admit the lower bound

$$\begin{aligned}
\langle z^k - \tilde{x}^k, Bx_{n+1}^k - w_{n+1}^k \rangle + \rho_k \langle \epsilon^k, Bx_{n+1}^k - B\tilde{x}^k \rangle \\
\geq \langle z^k - \tilde{x}^k, Bx_{n+1}^k - w_{n+1}^k \rangle - \underbrace{\rho_k \|\epsilon^k\| \|Bx_{n+1}^k - B\tilde{x}^k\|}_{\text{first part of } q_k'}.
\end{aligned}$$

Considering the first term on right-hand side of this bound, we also have

$$\begin{aligned}
\langle z^k - \tilde{x}^k, Bx_{n+1}^k - w_{n+1}^k \rangle &= \langle z^k - \tilde{x}^k, B\tilde{x}^k - w_{n+1}^k \rangle + \langle z^k - \tilde{x}^k, Bx_{n+1}^k - B\tilde{x}^k \rangle \\
&\geq \langle z^k - \tilde{x}^k, B\tilde{x}^k - w_{n+1}^k \rangle - \frac{d}{2} \|z^k - \tilde{x}^k\|^2 - \underbrace{\frac{1}{2d} \|B\tilde{x}^k - Bx_{n+1}^k\|^2}_{\text{second part of } q_k'}
\end{aligned}$$

for any $d > 0$, using Young's inequality. Finally, for the first two terms of the right-hand side of the above relation, we may write

$$\begin{aligned}
\langle z^k - \tilde{x}^k, B\tilde{x}^k - w_{n+1}^k \rangle &- \frac{d}{2} \|z^k - \tilde{x}^k\|^2 \\
&= \langle z^k - \tilde{x}^k, Bz^k - w_{n+1}^k \rangle + \langle z^k - \tilde{x}^k, B\tilde{x}^k - Bz^k \rangle - \frac{d}{2} \|z^k - \tilde{x}^k\|^2 \\
&\geq \underbrace{(\rho_k^{-1} - d/2) \|z^k - \tilde{x}^k\|^2 - \|z^k - \tilde{x}^k\| \|B\tilde{x}^k - Bz^k\|}_{q_k},
\end{aligned}$$

where in the final inequality we use the Cauchy-Schwartz inequality and substitute $Bz^k - w_{n+1}^k = \rho_k^{-1}(z^k - \tilde{x}^k)$, from the definition of $\tilde{x}^k$ in (80). We have now accounted for all the terms appearing in (79).

The recursion (79) is analogous to equation (F.7) on page 24 of Hsieh et al. (2020) and provides the starting point for the local convergence analysis. The next step would be to derive an analog of Theorem F.1. of Hsieh et al. (2020) using (79). The following translation to the notation of Theorem F.1. could be used (note that Hsieh et al. (2020) uses $t$ for iteration counter):

$$D_k = \|p^k - p^*\|^2,$$
$$\zeta_k = c_2 \alpha_k \rho_k (T'_k + l_k) + c_3 \alpha_k q_k,$$
$$\xi_k = -c_2 \alpha_k \rho_k r_k - c_3 \alpha_k r'_k,$$
$$\chi_k = c_1 \alpha_k^2 (\|e^k\|^2 + \|\epsilon^k\|^2 + \|p^k - p^*\|^2 + c_4) + c_5 \alpha_k q'_k,$$

and the event $E_\infty^\rho$ is translated to

$$E_\infty^\rho = \left\{ x_{n+1}^k \in \mathbb{B}_r(z^*), \tilde{x}^k \in \mathbb{B}_{\rho r}(z^*), p^k \in \mathbb{B}_{\rho r}(p^*) \text{ for all } k = 1, 2, \ldots \right\}.$$

An analog of Theorem 2 of Hsieh et al. (2020) could then be developed based on this result.

