# OpenReview forum: "Stochastic Projective Splitting: Solving Saddle-Point Problems with Multiple Regularizers"
_ICLR.cc/2022/Conference — ICLR 2022 Submitted_

### Official Review · Reviewer_vqpx · 2021-11-02

**Correctness:** 4
**Technical Novelty And Significance:** 3
**Empirical Novelty And Significance:** 2
**Recommendation:** 5
**Confidence:** 4

**Main Review:**

This manuscript is ambitious in tackling the problem of multiple operators with a stochastic oracle. This can be seen as a natural generalization of stochastic gradient algorithms for more than one regularizers to the setting monotone inclusion, and can find potential applications in min-max problems and so on.  The motivation is more or less just replacing the deterministic oracle in Johnstone & Eckstein (2020b) with a stochastic one and applying a different stepsize scheduling, but surely this is of importance in large-scale machine learning applications such that accessing all data points could be prohibitively expensive.  However, with the given stepsize scheduling, although being able to achieve optimality asymptotically by driving a certain proxy measure to  zero, the proposed algorithm does not honor the specific structure induced by the resolvents, and therefore can fail to achieve the original purpose of adding multiple operators.

My detailed comments are as follows.

1. In comparison with the product space reformulation, a clear issue is that multiple existing approaches generate iterates that honor the structure induced by the resolvents of the maximally monotone operators because the final step is the resolvent, while the proposed algorithm does not. In the context of optimization, this means that the specific structure (like sparsity, boundedness, low rank) enforced by the proximal operators will not be kept by the proposed algorithm. This is problematic from the application point of view (although probably not easily visible from the monotone inclusion angle), as usually the regularizers are added exactly because the users need specific structures in the final output.

2. The setting is interesting, but it looks like the authors did not really find a relevant application in practice and therefore needed to invent one of their own and therefore ignored a key part of such a regularizer. As mentioned above, addition of a regularization term is usually for the purpose to enforce or promote a desired structure in the solution, and for the case of the L1 norm, the purpose is usually sparsity, as mentioned by the authors. Therefore, an important aspect to show is the sparsity level and how it changes with time for these algorithms being compared. I can actually imagine from the algorithm forms that probably the proposed algorithm does not generate any sparsity, while Tseng's method, FRB, and FRB-VR will provide stable sparsity levels. This greatly reduces the usefulness of the proposed algorithm.

3. Given the slow convergence and the issue of not honoring the structure from the resolvents, I wonder if one could just apply (stochastic) subgradient methods or ideas similar to that, at least to a narrower range of problems that still include most the interesting cases. In such a case, likely the convergence speed will not be too different, thus from a theoretical angle, the proposed algorithm is probably not of very great contribution.

4. Another concern I have with the experiment is that it seems like the proposed algorithm is fast only in the beginning, and after reaching a mild precision, the optimality measure stalls. From the result on real-sim, it seems like that deterministic algorithms are superior and the proposed method is beneficial only at the beginning for going through more iterations within the same amount of time. If the algorithm is more useful in large-scale problems, for presentation purposes, the authors should indeed present more results on larger datasets and probably skip small scale ones like the one for real-sim.

5. Moreover, although the authors argued in the appendix that the product space formulation can lead to a much smaller step size that slows the convergence, it does not seem so in the comparison between deterministic PS and FRB or Tseng's method. It seems like that the deterministic PS is faster only in one case, thus diminishing the claimed contributions.

6. Judging from the tickers, it also seems to me that some of the advantages of the proposed algorithms is actually from the better time efficiency, in comparison with other methods. (Interestingly, usually one would expect deterministic methods to run faster because of the memory access pattern and less for-loops to be called, and the results presented are quite counterintuitive to me.) If the comparison is in terms of epochs (so that implementation difference can be excluded), I wonder if the outcome would be much different.

7. I also find the organization and detailed description quite unsatisfactory. It seems that the length limit of the main text of ICLR is too short for this manuscript to fully articulate, and I'd therefore suggest the authors to consider a journal or a different conference with a longer length limit (such as ICML or AISTATS that uses a two-column format). In particular, the authors spent quite a length in introducing fundamentals that might not be directly relevant to the rest parts of the main text and are actually well-known to most readers interested in this topic (those who decided to check the paper after viewing the title and abstract). On the other hand, key issues like how existing algorithms work, what the experiment is about, and how the residuals are estimated are all skipped in the main text. This essentially requires everyone to go back and forth between the appendices and the main text, which kind of violates the meaning of the appendices.
In essence, appendices are for things that do not affect the reader's grasp of the whole story, but clearly the authors are simply treating the appendices as places to implicitly increase the length of the main text and the readers are unable to skip any of the appendices without losing understanding of the main idea.


**Summary Of The Paper:**

This manuscript proposes a stochastic algorithm for monotone inclusion with more than two operators, where one is Lipschitz and the rest are maximally monotone and possibly set-valued.  Under standard noise conditions for a stochastic estimation for the Lipschitz operator, the authors showed almost sure convergence for step sizes satisfying some mild conditions, and provided convergence rates of O(k^{-1/4}) for an optimality measure if the number k is pre-specified.

**Summary Of The Review:**

The problem considered is of interest, but the proposed algorithm does not fully address the need of the applications. The manuscript presentation also has a great room for improvement. The current organization seems to be incomplete given the length of the main paper.

After reading the revision and the responses of the authors, and discussing with the authors, my recommendation moves to weak rejection. The idea is indeed interesting, but there are still unclear parts in the text and organization, and the motivation for the algorithm seems more like finding extensions for an existing work, instead of a real need for the considered problem.

---

> ### Author Response · Authors · 2021-11-11
> **Thanks for the review (reply part 1)**
>
> We thank the reviewer for their detailed report.
>
> >However, with the given stepsize scheduling, although being able to achieve optimality asymptotically by driving a certain proxy measure to zero, the proposed algorithm does not honor the specific structure induced by the resolvents, and therefore can fail to achieve the original purpose of adding multiple operators.
>
> We respectfully disagree with the reviewer.
>
> We understand the reviewers concern. Splitting algorithms utilize the proximal operators of nonsmooth functions. An appealing aspect of these algorithms is that the output of a prox operation will typically exhibit the structural property imposed by the regularizer. For example, for $x'=\text{prox}_{\rho||x||_1}(x)$, $x'$ will be *sparser* than the input $x$. For an algorithm utilizing this prox, the output of the prox will tend to be sparse after a small number of iterations for appropriate stepsizes and regularization constants.
>
> The reviewer claims that our method does not have this property. This is not true. Consider Line 4 of Algorithm 1 (our algorithm):
> $$
> x_i^k = J_{\tau A_i}(z^k+\tau w_i^k).
> $$
>
>  Each $x_i^k$ is the output of a resolvent calculation. In the simplest case of convex optimization, a resolvent is just a proximal operator, and each $x_i^k$ will exhibit the structural properties imposed by the corresponding regularizer with subdifferential $A_i$.
>
> Further note that our general convergence result applies to $x_i^k$ for $i=1,\ldots n$ just as it does for $z^k$. That is, $x_i^k\to z^*$ for a some solution $z^*$. This can be shown by noting that $z^k$ and $w_i^k$ both converge (a.s.), therefore $x_i^k = J_{\tau A_i}(z^k+\tau w_i^k)$ converges (a.s.). However, along the subsequence considered in (52) in Section C.9, $x_i^k-z^k\to 0$, therefore $x_i^k$ shares the same limit as $z^k$ (a.s.). We have added this as a remark at the end of Section C.9. Thanks to the reviewer for helping clarify this important point.
>
> It is worth mentioning that there is no way for *any* algorithm to exhibit/honor the structure of all resolvents simultaneously at all iterations. In general, this can only be done asymptotically. That is, in the limit, $z^k$ and $x_i^k$ all converge to a solution that will exhibit the structures imposed by all the regularizers. But for intermediate iterates, $x_i^k$ will only exactly exhibit the structure of the $i$th resolvent. This is not a flaw of our method but is true of all splitting methods. To see this, consider the case where the resolvents are projections onto convex sets $C_1,\ldots C_n$. Then, to "honor" resolvent $i$ is to produce a point in set $C_i$. But, if one could produce a point that simultaneously honors all the resolvents then one has produced a point in the intersection of $n$ convex sets in a finite number of iterations. But this is impossible in general as the convex feasibility problem can only be solved asymptotically by splitting methods.
>
> When there are multiple regularizers, a product-space application of a forward-backward-class method such as Tseng's algorithm will behave in the same manner:  the resolvent operations are applied in parallel, producing different subvectors of the next iterate.  Each subvector may "honor" the desired structure of the corresponding resolvent, but only one of them.
>
>  In summary, with just one regularizer, our method honors the structure of the nonsmooth operator if one focuses on the $x_1^k$ iterates instead of the $z^k$ iterates.  When there is more than one regularizer, it honors them in essentially the same manner as any splitting method using a product-space formulation if one considers the $x_i^k$ iterates (that is, individually as separate vectors but not jointly in a single vector).

---

> ### Author Response · Authors · 2021-11-11
> **Reply part 2**
>
> >Given the slow convergence and the issue of not honoring the structure from the resolvents, I wonder if one could just apply (stochastic) subgradient methods or ideas similar to that, at least to a narrower range of problems that still include most the interesting cases. In such a case, likely the convergence speed will not be too different, thus from a theoretical angle, the proposed algorithm is probably not of very great contribution.
>
> For the pure convex optimization case of our problem, it is possible to apply stochastic subgradient methods, although the performance is likely to be poor. Further, we hope we addressed the reviewer's concerns regarding whether our method "honors regularizers" above.
>
> However, for the general monotone inclusion case, including saddlepoint problems, the reviewer's suggestion has a serious flaw: there is no way to extend the subgradient method to a similar method for set-valued maximal montone operators. Let us explain what we mean.
>
> Consider a simplified case of our problem where there is only one set-valued operator and we are solving
> $$
> 0\in A(z).
> $$
> The extension of the subgradient method to this problem would be the following scheme:
> $$
> x^{k+1} = x^k - \rho_k a_k:\quad a_k\in A(x^k).
> $$
> Indeed if $A(z)=\partial f(z)$, then one can obtain ergodic function convergence rates for this scheme so long as the subgradients are bounded. However, for the more general inclusion problem, we are not aware of *any* convergence results for this scheme, and we believe such results are impossible without additional assumptions on $A$. The issue is that for the subgradient method we normally define convergence results in terms of *function values*, but for monotone inclusions, there is no function value, and one needs to study either $||a_k||$ or $||x^k-x^*||$ for a solution $x^*$. Typically, you need some additional assumption to derive convergence, such as strong monotonicity or cocoercivity of $A$.
>
> There are certain extensions of the extragradient method that weaken the *Lipschitz* assumption and allow for mere continuity. However, as far as we are aware, there are no such extensions that allow for arbitrary discontinuous or even set-valued operators such as we do.
>
>
> >Another concern I have with the experiment is that it seems like the proposed algorithm is fast only in the beginning, and after reaching a mild precision, the optimality measure stalls. From the result on real-sim, it seems like that deterministic algorithms are superior and the proposed method is beneficial only at the beginning for going through more iterations within the same amount of time. If the algorithm is more useful in large-scale problems, for presentation purposes, the authors should indeed present more results on larger datasets and probably skip small scale ones like the one for real-sim.
>
> We would argue that most stochastic algorithms are mainly beneficial in the early iterations. If one wants a high accuracy solution, then they should switch to a more accurate solver for fine-tuning. This is as much true for SGD as it is for our proposed method. For ML problems, high accuracy solutions are rarely required, since the empirical risk minimization problem only approximates the underlying prediction problem. Thus it is rarely necessary to switch to a more accurate solver.
>
> We included real-sim (a smaller problem) to show a diversity of problem sizes. It is true that our method does not perform better than the deterministic methods on this problem. We include it, in the interests of openness, as an example where a stochastic method may not be the best approach.
>
> >Moreover, although the authors argued in the appendix that the product space formulation can lead to a much smaller step size that slows the convergence, it does not seem so in the comparison between deterministic PS and FRB or Tseng's method. It seems like that the deterministic PS is faster only in one case, thus diminishing the claimed contributions.
>
> Our claim regarding small stepsizes was only relevent to the stochastic variant of Tseng, not to the deterministic variants, which can use constant stepsizes. The slowness of deterministic PS may be caused by it needing to compute a full gradient at each iteration - a drawback it has in common with the other determistic methods.

---

> ### Author Response · Authors · 2021-11-11
> **Reply part 3 (Final)**
>
> >Judging from the tickers, it also seems to me that some of the advantages of the proposed algorithms is actually from the better time efficiency, in comparison with other methods. (Interestingly, usually one would expect deterministic methods to run faster because of the memory access pattern and less for-loops to be called, and the results presented are quite counterintuitive to me.) If the comparison is in terms of epochs (so that implementation difference can be excluded), I wonder if the outcome would be much different.
>
> The reason we think our stochastic method is outperforming the deterministic variants is because it takes less time to compute a stochastic gradient, and yet similar quality search information is provided. This allows the stochastic methods to perform more iterations in the same time and hence to converge more quickly.
>
>
> We have included a comparison in terms of epochs in the appendix (Section I). It actually shows an even bigger benefit for the stochastic methods, since they have to process much less data in order to make progress than the deterministic batch methods. The difference is not as dramatic in the plots versus time, perhaps for some of the reasons you hinted at, such as multithreading, memory access patterns, overhead in the added iterations for the stochastic methods etc. These benefit the deterministic methods.

---

> ### Comment · Reviewer_vqpx · 2021-11-17
> **Thanks for the responses**
>
> I'd like to thank the authors for their detailed response. Some of my concerns are indeed well-explained in their response, especially regarding the convergence of $x^k$ and honoring the structures induced by the resolvents.
> The additional figures also resolved my concern about implementation difference.
> I am therefore updating my score to 5.
> The following issues are further derived from the concerns.
> 1. If $x^k$ are those that honor the structures and still possesses convergence guarantees, I would consider it more appropriate to report results of such intermediate variables, and report of the sparsity change is still in my view necessary.
> 2. The claim of not needing a higher solution precision is not very convincing with the current presentation. This is probably the case when the presented result is the objective in minimization, if all iterates obey the constraints. However, in this case, how the residual relates to the original objective is in question to me. How the iterates violate the constraints, how the objective is minimized, and how far the sparsity (even in $x$) is from that of the point of convergence (or at least how the sparsity changes or stablizes) are probably not directly relevant to the presented metric.
> 3. My concern about the experiment is not that real-sim shouldn't be presented, but that I think only 2 figures supporting the performance of the algorithm is not sufficient. I still hope at least more data sets can be tested upon.
> 4. Could you further deliberate on  the part that the small step size criticism is only for S-Tseng? In the current form it looks like S-Tseng also takes a fixed step size. If this is not the case, then details in both the algorithm description and the experiment setting should be given. Moreover, if the criticism is for only one approach, then why the whole subsection poses as if that's a drawback of all approaches using this strategy?

---

> > ### Author Response · Authors · 2021-11-18
> > **Thanks for the Reply**
> >
> > We thank the reviewer for taking the time to engage with us and again improving the quality of our paper.
> >
> > >1.  If  $x_k$  are those that honor the structures and still possesses convergence guarantees, I would consider it more appropriate to report results of such intermediate variables, and report of the sparsity change is still in my view necessary.
> >
> > The convergence rates we derive for the residual can also be extended to cover residuals related to each $x_i^k$ for $i=1,\ldots,n$. To see this, consider the residual in (65). Replace $z^k$ with $x_j^k$ for some $j=1,\ldots,n$ and call it $R_k^j$, the residual for $x_j^k$. Then $R_k^j$ can be shown to have the same rate as our main residual $G_k$ by following a similar argument to the upper bound derived for $R_k$  directly below (65).  We have added this as a comment below (65). Thank you for clarifying this point.
> >
> > We have included plots of the number of nonzeros versus running time for each method in Appendix I. One can see that our method does indeed produce sparse iterates for two of the problems. For the other, none of the methods are producing sparse iterates, probably because the scaling of the $L_1$ term needs to be increased if one wants a sparse solution.
> >
> > Please note that, in our opinion, focusing on sparsity is somewhat "unbalanced".  The test problem requires a saddle point condition coupled with several convex constraints, along with the sparsity-inducing $L_1$ term. We believe that it is better to use a comprehensive error metric integrating all these factors to assess algorithm performance (see our answer to question 2 below).  That one algorithm produces a sparser solution than another may not be very meaningful if it is farther from satisfying the constraints, for example.  It might also be misleading if the actual solution of the problem is not very sparse, as could easily occur depending on the  scaling of the $L_1$ term (in our case, the value of $c$).
> >
> > >2.  The claim of not needing a higher solution precision is not very convincing with the current presentation. This is probably the case when the presented result is the objective in minimization, if all iterates obey the constraints. However, in this case, how the residual relates to the original objective is in question to me. How the iterates violate the constraints, how the objective is minimized, and how far the sparsity (even in  $x$) is from that of the point of convergence (or at least how the sparsity changes or stablizes) are probably not directly relevant to the presented metric.
> >
> > We respectfully disagree with the reviewer. The presented metric does encompass constraint violation, sparsity (insofar as it causes the regularizer to differ from it's converged value), and the objective value. This is because, as shown in Lemma 1, the metric is zero *if and only if* $z^k$ solves the monotone inclusion and thus solves the min-max game or optimization problem under study. (As mentioned above, this residual can be extended to $x_i^k$ for $i=1,\ldots,n$.) This is why the residual is worth studying; it  represents a complete summary all of the various ingredients of the problem: constraints, nonsmooth regularizers, and smooth parts of the objective.
> >
> > Specific properties of the solution can be inferred from particular terms making up the residual $G_k$.  For instance, in our computational example $A_1$ enforced the constraints, while $A_2$ represents the $L_1$ regularizer.  The distance from contraint satisfaction is essentially measured by the $||z^k - x_1^k||$ term in $G_k$, while the difference from the stabilized point of sparsity is essentially measured by the $||z^k - x_2^k||$ term.  The $G_k$ terms involving $w_k$ essentially measure how close the entire ensemble $(z^k,w_1^k,...,w_n^k,w_{n+1}^k)$ is from forming a consistent systems of vectors certifying solution optimality.
> >
> > Another thing to note is that for games the objective value of a candidate solution alone does not tell you much about suboptimality. This is because the Nash equilibrium value of the game may be *higher* than the current value. This differs from minimization where the optimal value is always *lower*, so a candidate solution must give an upper bound. This is why for games, in order to compare different algorithms, we must turn to more integrative measures of solution quality, such as our residual.
> >
> > >3.  My concern about the experiment is not that real-sim shouldn't be presented, but that I think only 2 figures supporting the performance of the algorithm is not sufficient. I still hope at least more data sets can be tested upon.
> >
> > We completely understand the reviewer's concern that more empirical study of our method is needed. However, given the length of the appendix, we would like to avoid adding even more pages. We do plan on more extensive empirical testing in the future, However we prefer that the current paper to mainly focus on establishing the theoretical underpinnings of the proposed method.

---

> > > ### Comment · Reviewer_vqpx · 2021-11-19
> > > **My take on the performance metric**
> > >
> > > > Please note that, in our opinion, focusing on sparsity is somewhat "unbalanced". The test problem requires a saddle point condition coupled with several convex constraints, along with the sparsity-inducing term. We believe that it is better to use a comprehensive error metric integrating all these factors to assess algorithm performance (see our answer to question 2 below). That one algorithm produces a sparser solution than another may not be very meaningful if it is farther from satisfying the constraints, for example. It might also be misleading if the actual solution of the problem is not very sparse, as could easily occur depending on the scaling of the
> > >  term (in our case, the value of $c$).
> > >
> > > I focused on the sparsity part because that is the newly added regularizer in comparison to existing experiments setting, as the authors pointed out in the manuscript. Therefore, if the actual solution is not sparse, this seems to me more like an artificial problem that doesn't possess actual meaning in applications but more some simulated study, and that would harm the contributions of this work (as it seems like no real applications of multiple operators is available). Surely the constraint violation part is also crucial, and sometimes more important than the sparsity, because in many cases the constraints are either violated or not, but sparsity penalty is a continuous function.
> > > From the theoretical aspect, I understand the reasoning and motivation of using the residual as the measure, but in applications this is where I think something more meaningful is needed.
> > > But since the proposed method indeed generates sparsity while others fail, if this is a good application, anyway I'd consider that extra result as a plus for the method despite the conclusion from the discussion here.
> > >
> > > > We respectfully disagree with the reviewer. The presented metric does encompass constraint violation, sparsity (insofar as it causes the regularizer to differ from it's converged value), and the objective value. This is because, as shown in Lemma 1, the metric is zero if and only if solves the monotone inclusion and thus solves the min-max game or optimization problem under study. (As mentioned above, this residual can be extended to for .) This is why the residual is worth studying; it represents a complete summary all of the various ingredients of the problem: constraints, nonsmooth regularizers, and smooth parts of the objective.
> > >
> > > The result of Lemma 1 doesn't indicate anything related to approximate results. For example, for nonsmooth convex optimization problems, the minimum subgradient is zero iff an optimum is found, but it's possible that we can find points arbitrarily close to  an optimum with the minimum subgradient significantly larger than 0 (like the above example of converging to (1,0) for $\\min |x_1 - 1| + |x_2|$). Thus, something a little bit more than Lemma 1, like $G_k$ also converges to $0$ for points converging to a solution to the problem with certain properties, is needed to validate the claims of the authors, in my humble opinion. And since this is a new measure proposed by the authors, that one doesn't need to decrease it to very small seems lacking sufficient evidence. In most ML applications, it is the case for things like the objective, because it is shown by many researchers as well as practitioners in various cases that lower objective might not increase the prediction performance. But for the newly proposed measure, I think the claim of the authors for not needing it to decrease to too low is rather ungrounded.

---

> > > > ### Author Response · Authors · 2021-11-20
> > > > **Further Discussion: sparsity**
> > > >
> > > > >I focused on the sparsity part because that is the newly added regularizer in comparison to existing experiments setting, as the authors pointed out in the manuscript. Therefore, if the actual solution is not sparse, this seems to me more like an artificial problem that doesn't possess actual meaning in applications but more some simulated study, and that would harm the contributions of this work (as it seems like no real applications of multiple operators is available). Surely the constraint violation part is also crucial, and sometimes more important than the sparsity, because in many cases the constraints are either violated or not, but sparsity penalty is a continuous function. From the theoretical aspect, I understand the reasoning and motivation of using the residual as the measure, but in applications this is where I think something more meaningful is needed. But since the proposed method indeed generates sparsity while others fail, if this is a good application, anyway I'd consider that extra result as a plus for the method despite the conclusion from the discussion here.
> > > >
> > > > Thank you for considering the sparsity aspect a plus.  It's true that we started with an algorithmic idea and constructed a problem class that it can handle while being challenging for other methods.  However, we believe that the construction of the problem class is reasonable and not completely artificial: it adds a plausibly desirable feature to a useful problem class that didn't have a straightforward way to have that new feature included before.  We would also like to point out that one reason that existing practical problem classes of this complexity may be hard to find is because good algorithms did not exist for them.

---

> > > > ### Author Response · Authors · 2021-11-20
> > > > **Further Discussion: residual**
> > > >
> > > > >The result of Lemma 1 doesn't indicate anything related to approximate results. For example, for nonsmooth convex optimization problems, the minimum subgradient is zero iff an optimum is found, but it's possible that we can find points arbitrarily close to an optimum with the minimum subgradient significantly larger than 0 (like the above example of converging to (1,0) for $min(|x_1−1|+|x_2|)$. Thus, something a little bit more than Lemma 1, like $G_k$ also converges to 0 for points converging to a solution to the problem with certain properties, is needed to validate the claims of the authors, in my humble opinion.
> > > >
> > > > For our method it holds that $G_k\to 0$ almost surely. This can be shown fairly easily and we have included this in Section C.10. Thank you for helping to clarify this point. Does this answer your query?
> > > >
> > > > Note also that in the $min(|x_1−1|+|x_2|)$ example given, a one-operator proximal-point scheme (which our operations on the $A_i$ generalize) would in fact eventually jump exactly to the solution after a finite number of iterations.
> > > >
> > > >
> > > > >And since this is a new measure proposed by the authors, that one doesn't need to decrease it to very small seems lacking sufficient evidence. In most ML applications, it is the case for things like the objective, because it is shown by many researchers as well as practitioners in various cases that lower objective might not increase the prediction performance. But for the newly proposed measure, I think the claim of the authors for not needing it to decrease to too low is rather ungrounded.
> > > >
> > > > Sorry, we did not mean to say that it is not necessary to drive $G_k$ to very small values. We just meant to say that it is typical for stochastic algorithms to exhibit their main empirical convergence advantages during the early iterations and then to have comparably slow asymptotic convergence rates. This is indeed what we observe with our method and the other stochastic methods in this study.
> > > >
> > > > You're right to say that the connection between our residual and the *practical* performance of the trained model on learning problems is unclear. Our residual provides a measure of accuracy for the min-max problem -- but we did not test the practical performance of the trained classifier. This is a fair point and something we should look into, however, we feel that this is best left for a follow-up work. For the problem we study, it is not simply a matter of measuring prediction accuracy, since the min-max formulation is intended to trade off accuracy against robustness to distribution shift. Hence a more elaborate experiment would need to be conducted. We do not think we have the space to include this in our paper which is already quite long and its primary focus is the development of new optimization techniques suitable for applications in ML.
> > > >
> > > > Finally, we want to point out that, while the specific form of our residual in the context of projective splitting is new, our residual is really just the norm of a particular point in the image of the monotone operator of which we are trying to find a zero. This is explained in Section F.2. So it is not strictly  speaking a newly  proposed measure. It is in fact a generalization of the notion of a subgradient of a convex nonsmooth function and has been used in other papers for different algorithms.  And we do show that it converges to zero almost surely.

---

> > ### Author Response · Authors · 2021-11-18
> > **Reply to response part 2 (final)**
> >
> > >4.  Could you further deliberate on the part that the small step size criticism is only for S-Tseng? In the current form it looks like S-Tseng also takes a fixed step size. If this is not the case, then details in both the algorithm description and the experiment setting should be given.
> >
> > S-Tseng uses decaying stepsizes. The specific parameters are given in Table 1 in Appendix I. We have noticed that the previous version of the submission was ambiguous in that the caption of Table 1 did not actually mention S-Tseng. We have fixed this in the revision.
> >
> > > Moreover, if the criticism is for only one approach, then why the whole subsection poses as if that's a drawback of all approaches using this strategy?
> >
> > This is a fair point. Our criticisms in that section (Appendix F.7) do indeed apply to the deterministic methods as well. To be clear, based on this section, one might expect *deterministic* projective splitting to outperform the product space methods, and yet it does not in the experiments. That being said, in several other papers, deterministic projective splitting does outperform the product-space methods (for example https://arxiv.org/abs/1902.09025). So we will keep the original criticisms, but we have added a caveat in Section F.7 that the performance of these methods is actually similar in our experiment. Thank you for helping to make this clear.

---

### Official Review · Reviewer_H9EN · 2021-11-02

**Correctness:** 3
**Technical Novelty And Significance:** 4
**Empirical Novelty And Significance:** 4
**Recommendation:** 6
**Confidence:** 4

**Main Review:**

The paper is well written and the main contributions are clear. To the best of my knowledge this is the first paper that provides a stochastic variant of the projected splitting algorithm for solving monotone problems with convergence guarantees.

Below i provide some pointers for further clarification and improved presentation.

I believe that the broader audience of the ML community is not very familiar with the monotone inclusion problems. I would suggest the authors to expand further in the updated version the section 2 of the background on monotone inclusions. Have a proper statement of how these problems can formulate classical problems appearing in ML. For example the authors mentioned maximal operator without a proper definition. they also refer the interested reader to Ryu and Boyd (2016) for proper definitions. I think include all the necessary information in the main paper will be make this work self-contained and could reach broader audience.

Also i notice that there are few places that the authors could do better job in the presentation.

1) Add a reference for the equation (4) on first order necessary conditions for solving Nash-equilibrium problems and provide a bit more details of what this means.

2) More details required to be given on how one obtains the update rule (8) by projection step for a half space. What is the close form expression of $\nabla \phi_k$ and how this is computed relative to $\cal{P}$.

3) In theorems 1 and 2 add all assumptions on problem 1. Are the $A_i(z)$ and $B(z)$ monotone and L-Lipschitz? do you assume something beyond this and the noise assumptions for the proof to go through?

4) In the intro the authors mentioned that their algorithm includes the update rule of the algorithms proposed in Hsieh et al. 2020 as special case. However when they SPS method presented no more details are provided. can you elaborate more on this? how the method can be obtained as special case and how the convergence guarantees of the proposed method (Theorem 2) can be used in the setting of Hsieh et al. 2020?

5) On experiments: I would suggest instead of only comparing running time to also include 3 more plots where the horizontal axis will be computational complexity (how many gradient/operator evaluations are required by each method). this would be a fair comparison as the implementations might vary and not be optimal between the algorithms (this will affect the running time but not the computational complexity).

Some Missing References:
Four recent papers on the convergence analysis of stochastic algorithms for solving stochastic games and stochastic variational inequalities (special cases of monotone problems):

[1] Loizou, Nicolas, Hugo Berard, Alexia Jolicoeur-Martineau, Pascal Vincent, Simon Lacoste-Julien, and Ioannis Mitliagkas. "Stochastic hamiltonian gradient methods for smooth games." In ICML 2020.

[2] Loizou, Nicolas, Hugo Berard, Gauthier Gidel, Ioannis Mitliagkas, and Simon Lacoste-Julien. "Stochastic Gradient Descent-Ascent and Consensus Optimization for Smooth Games: Convergence Analysis under Expected Co-coercivity." NeurIPS 2021 (to appear).

[3] Mishchenko, Konstantin, Dmitry Kovalev, Egor Shulgin, Peter Richtárik, and Yura Malitsky. "Revisiting stochastic extragradient." In AISTATS 2020.

[4] Li, Chris Junchi, Yaodong Yu, Nicolas Loizou, Gauthier Gidel, Yi Ma, Nicolas Le Roux, and Michael I. Jordan. "On the convergence of stochastic extragradient for bilinear games with restarted iteration averaging." arXiv preprint arXiv:2107.00464 (2021).


**Summary Of The Paper:**

The paper presents a stochastic variant of the projective splitting family of algorithms for solving monotone inclusion problems. In particular, it explains how the monotone inclusion problems can formulate several popular machine learning tasks like min-max optimization problems, game formulation and variational inequality problems and provides convergence guarantees of the proposed algorithm for solving monotone problems.


**Summary Of The Review:**

As i mentioned in my main review i find that the paper is well written and the main contributions are clear. To the best of my knowledge this is the first paper that provides a stochastic variant of the projected splitting algorithm for solving monotone problems with convergence guarantees.

---

> ### Author Response · Authors · 2021-11-11
> **Thanks for the review**
>
> We thank the reviewer for their detailed response.
>
> >I believe that the broader audience of the ML community is not very familiar with the monotone inclusion problems. I would suggest the authors to expand further in the updated version the section 2 of the background on monotone inclusions. Have a proper statement of how these problems can formulate classical problems appearing in ML. For example the authors mentioned maximal operator without a proper definition. they also refer the interested reader to Ryu and Boyd (2016) for proper definitions. I think include all the necessary information in the main paper will be make this work self-contained and could reach broader audience.
>
> The definition of *maximal* monotone is already in the original version, in the paragraph entitled *Fundamentals* on page 3. While we appreciate the reviewer's concern, we believe there is already a lot of introductory material in Section 2. We show how to connect monotone inclusion problems to both optimization problems and games. Since we are already out of space, we decline to expand this section.
>
> >... Add a reference for the equation (4) on first order necessary conditions for solving Nash-equilibrium problems and provide a bit more details of what this means.
>
> (4) is derived by computing the first-order necessary conditions for the two optimization problems in (3). We added a sentence to clarify this. Thank you for the suggestion.
>
>
> >2.  More details required to be given on how one obtains the update rule (8) by projection step for a half space. What is the close form expression of  $\nabla\varphi_k$  and how this is computed relative to  P.
>
>   We included the closed form expression for $\nabla\varphi_k$ in the revision. Deriving the update rule is a straightforward Lagrange-multiplier exercise in computing the hyperplane projection step so we did not include it.
>
> >3.  In theorems 1 and 2 add all assumptions on problem 1. Are the  $A_i(z)$  and  $B(z)$  monotone and L-Lipschitz? do you assume something beyond this and the noise assumptions for the proof to go through?
>
>   We explicitly added the assumptions to Theorem 1 in the revision. All the $A_i$ and $B$ are maximal monotone, however *only* $B$ is Lipschitz. Nothing else is assumed.
>
> >4.  In the intro the authors mentioned that their algorithm includes the update rule of the algorithms proposed in Hsieh et al. 2020 as special case. However when they SPS method presented no more details are provided. can you elaborate more on this? how the method can be obtained as special case and how the convergence guarantees of the proposed method (Theorem 2) can be used in the setting of Hsieh et al. 2020?
>
> If you set $n=1$, then by line 10, $w_1^{k+1}=w_1^k$. By the initialization condition, $w_1^0=0$, hence $w_1^k=0$. The updates on line 7 and 9 become
> $$
> x_1^k = z^k - \rho_k (B(z^k)+\epsilon^k)
> $$
> and
> $$
> z^{k+1} = z^k - \alpha_k(B(x_1^k)+e^k).
> $$
> This is the same form as Hsieh et al. 2020 and further our stepsize constraints in (12) are the same as theirs.
>
>
> >On experiments: I would suggest instead of only comparing running time to also include 3 more plots where the horizontal axis will be computational complexity (how many gradient/operator evaluations are required by each method). this would be a fair comparison as the implementations might vary and not be optimal between the algorithms (this will affect the running time but not the computational complexity).
>
> We have included a plot versus data passes in the appendix. This is a fair comparison because all methods have similar complexities beyond the matrix multiplications they perform in accessing the dataset. The deterministic methods only ever access the entire data matrix, while the stochastic methods subsample slices.
>
>
> >Some Missing References: Four recent papers on the convergence analysis of stochastic algorithms for solving stochastic games and stochastic variational inequalities (special cases of monotone problems):
> >
> >[1] Loizou, Nicolas, Hugo Berard, Alexia Jolicoeur-Martineau, Pascal Vincent, Simon Lacoste-Julien, and Ioannis Mitliagkas. "Stochastic hamiltonian gradient methods for smooth games." In ICML 2020.
> >
> >[2] Loizou, Nicolas, Hugo Berard, Gauthier Gidel, Ioannis Mitliagkas, and Simon Lacoste-Julien. "Stochastic Gradient Descent-Ascent and Consensus Optimization for Smooth Games: Convergence Analysis under Expected Co-coercivity." NeurIPS 2021 (to appear).
> >
> >[3] Mishchenko, Konstantin, Dmitry Kovalev, Egor Shulgin, Peter Richtárik, and Yura Malitsky. "Revisiting stochastic extragradient." In AISTATS 2020.
> >
> >[4] Li, Chris Junchi, Yaodong Yu, Nicolas Loizou, Gauthier Gidel, Yi Ma, Nicolas Le Roux, and Michael I. Jordan. "On the convergence of stochastic extragradient for bilinear games with restarted iteration averaging." arXiv preprint arXiv:2107.00464 (2021).
>
> These references have been added. Thank you for the suggestions.

---

> > ### Comment · Reviewer_H9EN · 2021-11-25
> > **Review Update**
> >
> > Thank you to the authors for providing further clarification to the raised points.
> > I have read the other reviews, the rebuttal, and browsed through the paper again.
> >
> > I have decided to maintain my original score.

---

### Official Review · Reviewer_LTCL · 2021-11-03

**Correctness:** 4
**Technical Novelty And Significance:** 2
**Empirical Novelty And Significance:** 3
**Recommendation:** 5
**Confidence:** 4

**Main Review:**

One of the drawbacks of this paper is that the main idea of the algorithm is somehow based on several works in literature. Specifically, the algorithm SPS is a stochastic variant of the recently proposed projective splitting (PS) [Johnstone & Eckstein, 2020b]. Also, the Double Stepsize Extragradient Method (DSEG) [Hsieh et al., 2020] is a special case of SPS when removing all regularizers and constraints. To address this issue, the authors discussed the comparisons of their results with other related works and explains the significance of this work lies in the general applicability of their convergence rate, which is one of the advantages of this work. When compared with the preexisting PS framework, SPS enables computational efficiency on large datasets. When compared with other stochastic methods on special cases of monotone inclusions, SPS enables a strong ability to deal with constraints and regularizations.

However, the theoretical convergence rate derived in this paper is no better than previous results when constrained to specific problems in other works. While the problem setting is indeed more general, I feel confused about the possible technical difficulties that have caused the $\mathcal{O}(K^{-1/4})$ convergence rate. In [Hsieh et al., 2020] they proved a $1/K^{1/3}$ last iterate convergence rate for the stochastic case, although Theorem 2 seems directly applicable to this setting, the convergence rate is $\mathcal{O}(K^{-1/4})$. From my viewpoint, a good convergence result under the general framework should not only be adaptive to the special cases but also obtain a sharp rate when specified to traditional settings.

With this consideration, the contribution of this piece of work is a little weak. Moreover, why does the experimental result seem to outperform previous works, when the theory does not? In the first figure the pink FRB-VR curve seems to be descending at the right boundary, is it possible that FRB-VR reaches better solutions under some circumstances?


**Summary Of The Paper:**

The paper focuses on the stochastic variant of the projective splitting (PS) algorithm. With a specific focus on monotone inclusion problems, the authors propose a novel separable algorithm featured by the ability to handle multiple constraints and non-smooth regularizers. Compared with similar approaches on variational inequality which is a special case of monotone inclusions, this paper uses a more direct error metric than the restricted gap function. Moreover, although with a slower convergence rate, this paper is the first discussion under the general discontinuous monotone inclusion case.

**Summary Of The Review:**

Overall, the paper is well written with each part of the results clearly explained and well supported. This is a highly interesting piece of work for the SPS problem, but the significance of its contribution is vague to me.

---

> ### Author Response · Authors · 2021-11-11
> **Thanks for the review**
>
> We thank the reviewer for their detailed report.
>
>
> > However, the theoretical convergence rate derived in this paper is no better than previous results when constrained to specific problems in other works. While the problem setting is indeed more general, I feel confused about the possible technical difficulties that have caused the $O(K^{−1/4})$ convergence rate. In [Hsieh et al., 2020] they proved a $1/K^{1/3}$ last iterate convergence rate for the stochastic case, although Theorem 2 seems directly applicable to this setting, the convergence rate is $O(K^{−1/4})$. From my viewpoint, a good convergence result under the general framework should not only be adaptive to the special cases but also obtain a sharp rate when specified to traditional settings.
>
> The $O(K^{-1/3})$ rate of [Hsieh et al. 2020] requires **strong monotonicity**, or at least partial strong monotonicity (i.e. an error bound). We do not. This is why our rate is worse. We can also obtain a $O(K^{-1/3})$ rate under appropriate conditions (Lipschitz + strong monotonicity).
>
> Ordinarily, when going to a more general setting, convergence rates will get worse, as the problem setting is larger and so there is more scope for "difficult" counterexamples to arise. One should not directly compare our rates with those of convex optimization, for instance, since our problem setting (monotone inclusions) is more general. Nevertheless, finding out whether our rate is tight, or could be improved either for our method or with another splitting method, is an important topic to study. Since our paper is already quite long, we defer this to future work.
>
> >Moreover, why does the experimental result seem to outperform previous works, when the theory does not?
>
> None of the other tested stochastic methods have known convergence rates for their residual. The other methods are only known to converge asymptotically or have convergence rates for the variational inequality gap function, which is not directly comparable to the residual studied in our paper.  So it is not strictly true to say that our method has no theoretical benefits here.
>
> >In the first figure the pink FRB-VR curve seems to be descending at the right boundary, is it possible that FRB-VR reaches better solutions under some circumstances?
>
> We agree that the right horizon for this plot did seem possibly misleading. We replotted it with the right horizon set to 200 sec (see the revision). It should now be clear that FRB-VR does not reach better solutions in this case. It is slower and does not plateau at values as good as our two methods. (Note that the colors have changed since the last plot for inscrutible Python reasons).

---

> ### Author Response · Authors · 2021-11-18
> **Any Further Comments?**
>
> We hope that reviewer LTCL has had a chance to read our response. Do you have any more questions?
>
> We hope you can understand that our rate does not meet those of [Hsieh et al. 2020] simply because we make less strict assumptions.
>
> We have also redrawn the first figure to better distinguish our performance from FRB-VR.

---

> ### Comment · Reviewer_LTCL · 2021-11-30
> **Thanks for the response**
>
> I thank the authors for a detailed response and their deliberate discussions with Reviewer H9EN and Reviewer vqpx. The authors have addressed part of my concerns, and I believe this paper does have distinctive aspects compared with previous literature. However, for the current version, the contribution of the proposed algorithm seems limited. At least the authors could consider adding part of the discussions to the manuscript. Therefore, I've decided to keep my score as it is.

---

> > ### Author Response · Authors · 2021-11-30
> > **Thanks for clarifying**
> >
> > We thank the reviewer for clarifying their position.
> >
> > We had thought that your primary concern about the paper was a slow convergence rate, which we have addressed. But it appears you are also concerned with the novelty of the work more generally. We guess that your claim of limited novelty is related to the following excerpt from your original comment:
> >
> > >One of the drawbacks of this paper is that the main idea of the algorithm is somehow based on several works in literature.
> >
> > We would only point out that **all** algorithms in optimization are based on previous works in some sense.
> >
> > For what it's worth, we stress that our analysis is a significant departure from previous projective splitting papers. The iterate convergence is based on stochastic Quasi-Fejer monotonicity, which no projective splitting paper has used before. The convergence rate analysis is completely novel - no previous projective splitting paper has studied this metric. And as you said in your original comment, our method has several benefits over others - stochastic updates and the easy ability to handle constraints/regularizers being two prime examples.
> >
> > Nevertheless, questions of novelty are subjective, and you're entitled to stick with your current position.

---

### Author Response · Authors · 2021-11-29
**Still waiting for Reviewer LTCL**

As of Nov 29 we still have not received a reply from Reviewer LTCL.

Does the reviewer have any other questions or comments? We believe we answered all of their concerns. If the reviewer has no other concerns, then we ask: why have they not reconsidered their score? If they have a reason not to increase their score, then they have a duty to tell us why.

We understand reviewers are under a lot of pressure with many obligations as well as their own work/life commitments. However, today is the last day of the discussion period and we are still patiently awaiting a response.

Thank you,

---

### Decision · Program_Chairs · 2022-01-20

**Decision:**

Reject

**Comment:**

The submission considers a stochastic variant of the projective splitting algorithm, with a focus on monotone inclusion problems, and it proposes a novel separable algorithm with the ability to handle multiple constraints and non-smooth regularizers.  All reviewers felt that there were merits to the submission and that the submission was borderline.  Public and non-public discussion concluded that the paper would be of greater value to the community if the suggestions of the reviewers and related issues were addressed.